# Structural insight into toxin secretion by contact-dependent growth inhibition transporters

Jeremy Guerin[1], Istvan Botos[1], Zijian Zhang[2], Karl Lundquist[2], James C Gumbart[2], Susan K Buchanan[1]*

[1]Laboratory of Molecular Biology, NIDDK, NIH, Bethesda, United States; [2]School of Physics, Georgia Institute of Technology, Atlanta, Georgia

**Abstract** Bacterial contact-dependent growth inhibition (CDI) systems use a type Vb secretion mechanism to export large CdiA toxins across the outer membrane by dedicated outer membrane transporters called CdiB. Here, we report the first crystal structures of two CdiB transporters from *Acinetobacter baumannii* and *Escherichia coli*. CdiB transporters adopt a TpsB fold, containing a 16-stranded transmembrane β-barrel connected to two periplasmic domains. The lumen of the CdiB pore is occluded by an N-terminal α-helix and the conserved extracellular loop 6; these two elements adopt different conformations in the structures. We identified a conserved DxxG motif located on strand β1 that connects loop 6 through different networks of interactions. Structural modifications of DxxG induce rearrangement of extracellular loops and alter interactions with the N-terminal α-helix, preparing the system for α-helix ejection. Using structural biology, functional assays, and molecular dynamics simulations, we show how the barrel pore is primed for CdiA toxin secretion.

*For correspondence:
susan.buchanan2@nih.gov

Competing interests: The authors declare that no competing interests exist.

## Introduction

In bacterial ecosystems, competition for limited nutrients can be a life or death battle. To fight for resources, some Gram-negative bacteria employ direct toxin exchange through a process known as Contact-Dependent growth Inhibition (CDI). This process was first described in *Escherichia coli* EC93, where a two-partner secretion system consisting of a CdiA toxin and a CdiB transporter was shown to inhibit other *E. coli* strains (*Aoki et al., 2005*). CdiB is an outer membrane transporter that releases its CdiA toxin to the cell surface. Once contact occurs, a toxin domain at the CdiA C-terminus is cleaved and imported into the target bacterium to inhibit growth. To prevent self-destruction, CDI systems also express an immunity protein, CdiI, which protects against CdiA toxins delivered from neighboring cells (*Figure 1—figure supplement 3*; *Aoki et al., 2005*; *Ruhe et al., 2018*; *Ruhe et al., 2017*).

CdiA and CdiB belong to the Two-Partner Secretion family of proteins (TPS; Type Vb secretion system). The core of a TPS system consists of two proteins called TpsA for the secreted proteins, and TpsB for their cognate transporters (*Guérin et al., 2017*). Like TpsA, CdiA toxins are predicted to fold into a β-helix, forming an elongated filament that extends several hundred angstroms from CdiB transporters (*Clantin et al., 2004*; *Ruhe et al., 2018*). CdiA proteins are synthesized in the cytoplasm and contain two N-terminal domains directing their secretion: a signal peptide and a TPS domain (*Figure 1—figure supplement 3*). After inner membrane translocation and signal peptide removal by the SEC machinery, the CdiA TPS domain interacts with the periplasmic domains of its cognate CdiB transporter (*Baud et al., 2014*; *Clantin et al., 2004*; *Delattre et al., 2011*; *Hodak et al., 2006*). At this point, translocation across the outer membrane is initiated and the rest of the protein is folded at the surface of the bacterium. Domain organization and folding during

secretion are still poorly understood, however a study using electron cryo-tomography suggests that secretion occurs in two distinct steps (*Ruhe et al., 2018*). In this model, the CdiA N-terminal half of the protein, including the TPS and FHA-1 domains, is secreted first and forms a 330 Å filament exposing the receptor-binding domain (RBD). The RBD recognizes a specific membrane receptor on the surface of a neighboring cell, triggering the second secretion step (*Ruhe et al., 2018*; *Ruhe et al., 2017*). The CdiA C-terminal half of the protein, which was still in the periplasm, is now released and exported to the cell surface. A short tyrosine- and proline-rich region and a second filamentous hemagglutinin domain (FHA-2) fold and then associate with the outer membrane of the target bacterium to deliver the C-terminal toxin. The toxin domain is cleaved from the rest of the CdiA protein by an unknown mechanism, and then released into the target cell.

CdiB proteins are members of the Omp85 superfamily. There are two functionally distinct protein classes in the Omp85 superfamily: BamA/TamA proteins that insert newly synthesized outer membrane proteins into the outer membrane, and TpsB proteins that secrete cognate protein substrates to the extracellular surface (*Figure 1—figure supplement 3*; *Gentle et al., 2004*; *Guérin et al., 2017*; *Heinz and Lithgow, 2014*). While there are several BamA, TamA, and BAM complex structures (*Bakelar et al., 2016*; *Gruss et al., 2013*; *Gu et al., 2016*; *Noinaj et al., 2013*), only one TpsB structure has been characterized: *Bordetella pertussis* FhaC (*Clantin et al., 2007*; *Maier et al., 2015*). This TpsB transporter secretes a major adhesin called filamentous haemagglutinin (FHA). BamA/TamA and TpsB proteins share a common fold with distinct features enabling either insertion into the membrane, or secretion across it. The two protein classes use a 16-stranded β-barrel to span the outer membrane, connected to a series of N-terminal periplasmic interaction modules called polypeptide-transport-associated (POTRA) domains. Inside the β-barrel lumen, extracellular loop 6 (L6) forms a 'lid-lock' through interactions between two essential signature motifs: (V/I)**R**G(Y/F) at the tip of L6 and (F/G)x**D**xG on strand β13 (*Gruss et al., 2013*; *Maier et al., 2015*; *Noinaj et al., 2013*). Mutagenesis experiments have shown that L6 is essential for activity but its precise function is unclear (*Guérin et al., 2015*; *Höhr et al., 2018*; *Leonard-Rivera and Misra, 2012*; *Rigel et al., 2013*). In addition, TpsB proteins contain an N-terminal α-helix (H1) inserted into the barrel pore that is not found in BamA/TamA proteins. H1 is connected to the first POTRA domain by a short periplasmic polypeptide; this linker has been shown to be essential for secretion in the FhaC/FHA system. Since H1 blocks the barrel pore in the resting conformation, it must be removed for secretion to occur (*Figure 1—figure supplement 3*; *Baud et al., 2014*; *Guérin et al., 2014*; *Maier et al., 2015*).

Here we report the first crystal structures of CdiB transporters from *A. baumannii* (ACICU) and *E. coli* (EC93). Two distinct conformations for H1 are observed within the β-barrel lumen. Using structure-based sequence alignment, we identified a conserved DxxG motif on strand β1 that is found in all TpsB transporters but not in BamA/TamA proteins. We show that the role of DxxG is to increase the flexibility of strand β1, which in turn affects the β1–β16 interface. We developed a secretion assay to show that CdiB transporters specifically secrete their cognate CdiA proteins and used this assay to analyze the functions of individual amino acids in CdiA secretion. Molecular dynamics simulations illustrate ejection of H1 from β-barrel lumen. Our results highlight conformational changes in the β-barrel domain that facilitate pore opening and secretion of the substrate.

## Results

### Two CdiB transporter structures

We determined two full-length crystal structures of CdiB transporters from *Acinetobacter baumannii* (strain ACICU) and *E. coli* (strain EC93), which will be referred to as CdiB[Ab] and CdiB[Ec]. CdiB[Ab] and CdiB[Ec] structures were built and refined to final resolutions of 2.4 Å and 2.6 Å, respectively (*Table 1*). Despite low sequence similarity (21% sequence identity), both structures adopt a common fold: an N-terminal α-helix (H1) is inserted into the lumen of the β-barrel and is connected by a ~ 20 residue linker to two periplasmic POTRA domains. The POTRA domains have a conserved βααββ fold and extend away from the β-barrel (*Figure 1*). The C-terminal β-barrel consists of 16 antiparallel β-strands organized as an oblique cylinder with cross-sectional dimensions of 35 Å x 25 Å. The longest β-strands (β5 to β8) form an extended β-sheet that may serve to anchor the CdiA substrate to initialize its folding (*Baud et al., 2014*; *Figure 1* and *Supplementary file 1*).

**Table 1.** Data collection and refinement statistics for CdiB$^{Ab}$ and CdiB$^{Ec}$.

| | CdiB Ab (Se) | CdiB Ab | CdiB Ec |
|---|---|---|---|
| **Data collection** | | | |
| λ (Å) | 0.979415 | 1.0 | 1.0 |
| Space group | P1 | P1 | P2$_1$2$_1$2 |
| a, b, c (Å) | 47 49.3 86.8 | 46.9 49.3 86.8 | 45.3 112.9 183.4 |
| α, β, γ (°) | 100.7 90.6 109.9 | 100.8 90.4 109.9 | 90 90 90 |
| Resolution (Å) | 50–2.6 | 50–2.4 | 50–2.6 |
| R$_{sym}$/R$_{merge}^{†*}$ | 0.1 (1.4) | 0.1 (1.3) | 0.1 (1.3) |
| I / σ (I)$^*$ | 13.6 (1.4) | 10.1 (1.3) | 13.9 (2.0) |
| CC (1/2) (%)$^*$ | 0.998 (0.688) | 0.99 (0.619) | 0.99 (0.9) |
| Completeness (%)$^*$ | 90.8 (90.0) | 97.0 (82.4) | 99.9 (100) |
| Ano Completeness (%)$^*$ | 90.1 (89.5) | | |
| Redundancy$^*$ | 7.7 (7.7) | 5.8 (5.6) | 13 (13.5) |
| **Refinement** | | | |
| Resolution (Å) | | 44–2.4 | 44–2.6 |
| No. reflections | | 27153 | 29889 |
| R$_{work}^{§}$/R$_{free}^{¶}$ | | 0.20/0.25 | 0.24/0.26 |
| **r.m.s. deviations** | | | |
| Bonds (Å) | | 0.003 | 0.002 |
| Angles (°) | | 0.61 | 0.58 |
| No. Protein atoms | | 4256 | 4130 |
| No. Ligand atoms | | 52 | 90 |
| No. Waters | | 55 | 13 |
| **B-factors (Å$^2$)** | | | |
| Wilson B | | 53.20 | 68.29 |
| Protein | | 59.6 | 79.1 |
| Ligands | | 69.1 | 81.2 |
| Waters | | 54.4 | 58.5 |
| **Ramachandran Analysis** | | | |
| Favored (%) | | 98.3 | 97.18 |
| Allowed (%) | | 1.7 | 2.82 |
| Outliers (%) | | 0 | 0 |
| **PDB code** | | 6WIL | 6WIM |

†$R_{sym} = \Sigma_{hkl,j} (|I_{hkl}-<I_{hkl}>|) / \Sigma_{hkl,j} I_{hkl}$, where $<I_{hkl}>$ is the average intensity for a set of j symmetry-related reflections and $I_{hkl}$ is the value of the intensity for a single reflection within a set of symmetry-related reflections.

§$R$ factor $= \Sigma_{hkl} (||F_o| - |F_c||) / \Sigma_{hkl}|F_o|$ where $F_o$ is the observed structure factor amplitude and $F_c$ is the calculated structure factor amplitude.

¶$R_{free} = \Sigma_{hkl,T} (||F_o| - |F_c||) / \Sigma_{hkl,T}|F_o|$, where a test set, T (5% of the data), is omitted from the refinement.

* Statistics for highest resolution shell shown in parentheses.

The extracellular surface of the β-barrel is composed of long loops in mostly extended conformations that allow access to the barrel pore and would facilitate CdiA secretion. In TpsB/CdiB proteins, extracellular loop 5 is usually longer than other extracellular loops; the entire loop 5 can be traced in CdiB$^{Ec}$ but electron density is missing for CdiB$^{Ab}$ (residues 389–411) and FhaC (residues 381–399; PDB: 4QKY). In CdiB$^{Ec}$, loop 5 is stabilized by contacts with loop 6 (R421) and two residues from strand β8 (R340 and T343) (*Figure 1—figure supplement 1*). Interestingly, this loop contains three

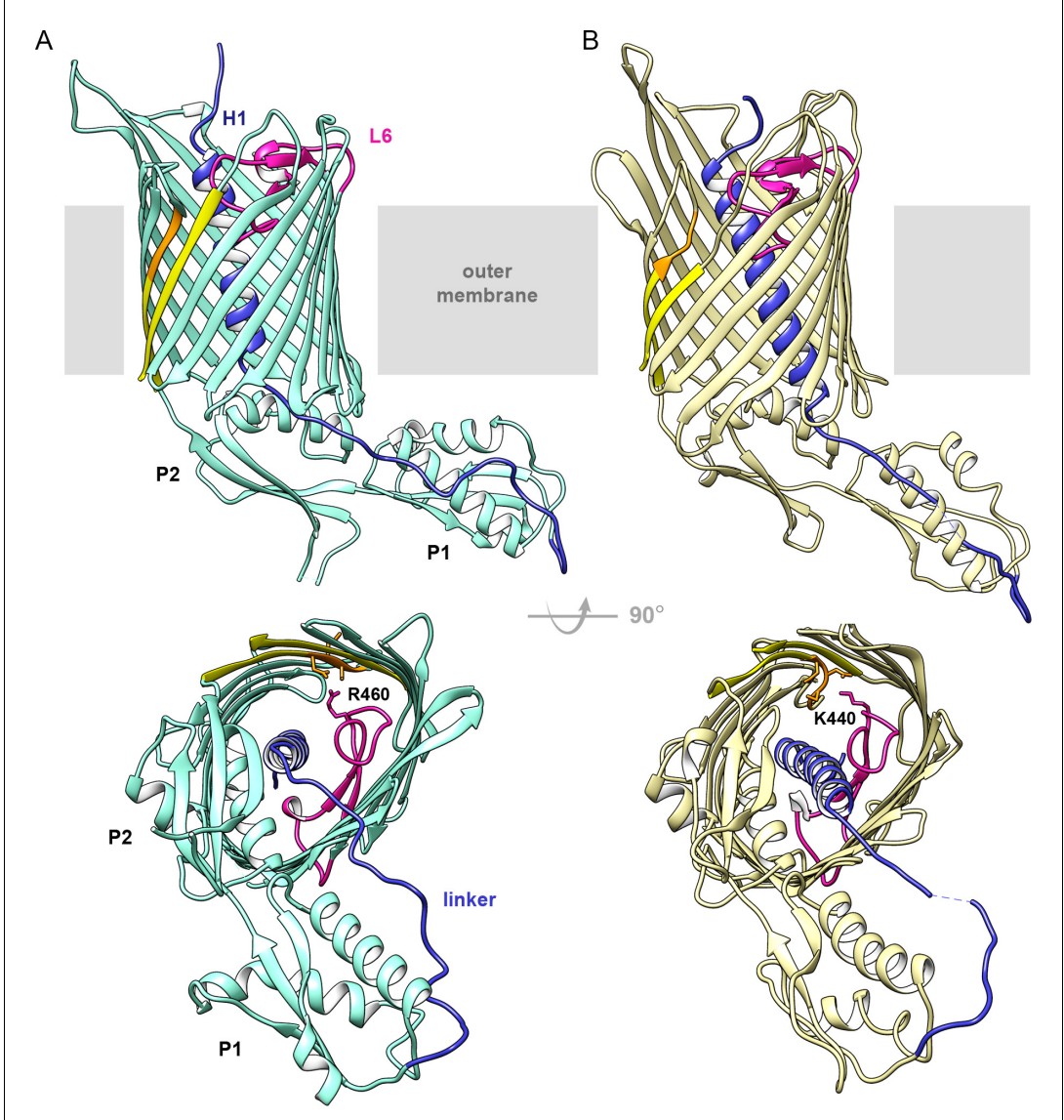

**Figure 1.** Structures of CdiB transporters. Membrane view (upper panel) and periplasmic view (lower panel) of (**A**) CdiB from *Acinetobacter baumannii* (CdiB[Ab]) in light teal and, (**B**) CdiB from *Escherichia coli* (CdiB[Ec]) in pale yellow. The first POTRA domain is indicated by P1, and the second POTRA domain, closer to the β-barrel, by P2. Inside the β-barrel, the N-terminal helix H1 is shown in blue and Loop 6 (L6) in magenta. The linker connecting H1 to P1 is also colored blue. The first and last β-strands from the β-barrel are shown in yellow, with the DxxG motif in orange. In the lower panels, selected sidechains from DxxG and L6 are shown as sticks, with the interacting loop 6 residue highlighted.

The online version of this article includes the following figure supplement(s) for figure 1:

**Figure supplement 1.** Positions of extracellular loops in the CdiB[Ec] structure.

**Figure supplement 2.** Network of interactions connecting the β-barrel, POTRA domains, and Linker.

**Figure supplement 3.** Biogenesis of CDI proteins.

hydrophobic residues (M370, W372, F373) that extend outward from the β-barrel toward the membrane, with sidechains potentially interacting with the bilayer lipopolysaccharides.

On the periplasmic side of the outer membrane, the two POTRA domains exhibit the conserved βααββ fold characteristic of Omp85 proteins (*Figure 1—figure supplement 2*). A network of interactions connects β-barrel-POTRA2 and the linker through conserved charged residues. A glutamate from helix α4 of POTRA2 (E179 in CdiB[Ab] and E168 in CdiB[Ec]) forms a salt bridge with an arginine and a lysine present on strands β6 and β7, respectively (R325-K330 in CdiB[Ab] and R309-K314 in CdiB[Ec]), while the N-terminal region of the linker (Y35 in CdiB[Ab] and S34-A35 in CdiB[Ec]) is also stabilized

by side chains from α4 (D175 for CdiB[Ab], and R161-E164 for CdiB[Ec]). Some of these interactions are also seen in the FhaC structure, and biochemical experiments have shown that the interactions between POTRA2 and the linker are essential for substrate recognition and secretion (*Baud et al., 2014*; *Delattre et al., 2011*; *Maier et al., 2015*). These conserved interactions emphasize the importance of the POTRA domains for TpsB function.

The linker connecting H1 to POTRA1 is well defined in the electron density maps. In CdiB[Ab], the linker is stabilized by a network of interactions with helices α2 and α4 from POTRA1 and POTRA2, respectively (*Figure 1—figure supplement 2*). In both CdiB structures, the C-terminal part of the linker is attached to POTRA1 by a disulfide bond between two cysteines, an interaction that is conserved in all CdiB transporters (and some TpsB transporters). While a network of interactions stabilizes the N-terminal linker region near the β-barrel lumen, the middle region of the linker appears more flexible, sharing fewer surface interactions with POTRA1 (especially for CdiB[Ec] where residues 42–43 are missing in the electron density). This position differs slightly between CdiB[Ab] and CdiB[Ec], confirming the biochemical and biophysical studies that have shown multiple linker conformations in the resting state (*Guérin et al., 2014*; *Maier et al., 2015*; *Figure 1*, *Figure 1—figure supplement 2*).

## Structural differences in helix H1

Both CdiB structures adopt the same overall architecture, with H1 and L6 occluding the interior of the β-barrel. However, H1 is positioned very differently in CdiB[Ab] and CdiB[Ec] (*Figure 2*, *Video 1*): the angle between H1 and the β-barrel is about 10° for CdiB[Ab], versus 25° for CdiB[Ec]. In addition, H1 sits higher in the barrel pore in CdiB[Ab], positioning its N-terminus closer to the extracellular surface. In this orientation, H1 interacts with the inner barrel wall using 12 H-bonds and three salt bridges, with no interactions to loop 6, for a total buried surface area of 1263 $Å^2$ (*Figure 2*, *Figure 2—source data 1*). In CdiB[Ec], H1 in sits 4.8 Å lower in the barrel pore, allowing it to form 5 H-bonds with loop 6, and 8 H-bonds with the barrel wall, for a total buried surface of 1385 $Å^2$.

To better understand the orientation of H1 in the β-barrel lumen, we generated two models using the targeted molecular dynamics method (TMD). The first model is based on the structure of CdiB[Ab], where the helix H1 is moved from its initial position toward the position of the CdiB[Ec] helix H1. The second model is based on the structure of CdiB[Ec], where the helix H1 is moved from its initial position toward the position of CdiB[Ab] helix H1. TMD was run for 30 ns, followed by 30 ns of H1 restrained to the new position, and then 120 ns of free equilibration (*Video 2*). For both models, analysis of RMSD during free equilibration reveal that H1 is stable in the new position and does not revert to the initial state (*Figure 2—figure supplement 1*, *Video 2*). This result suggests that H1 can adopt either orientation in both systems.

Our results on the position of H1 in CdiB agree with previous studies that have reported high flexibility for H1 in the TpsB transporter FhaC. Acting as a plug domain, H1 exists in different states in the resting conformation and can undergo a large conformational change to fully open the β-barrel pore during secretion (*Baud et al., 2014*; *Guérin et al., 2014*). The structural variations observed in our CdiB structures illustrate at least two conformations that include different interactions with the interior of the β-barrel, loop 6 and extracellular loops.

## Extracellular loop 6 and the DxxG motif on strand β1

Like H1, L6 sits in the interior of the β-barrel and partially occludes it. This conformation is maintained by a conserved salt bridge between an arginine on the tip of L6 (from the (V/I)**R**G(F/Y) motif) and an aspartate on strand β13 ((F/G)x**D**xG motif), an interaction observed in all Omp85 structures (*Gruss et al., 2013*; *Gu et al., 2016*; *Maier et al., 2015*; *Noinaj et al., 2013*). In addition, in the CdiB[Ab] and CdiB[Ec] structures, the arginine is also stabilized by a glutamate from β12 (E483 and E455 respectively) (*Figure 3A and B*). This position inside the β-barrel lumen allows loop 6 sidechains to point toward the β1–β16 interface. Loop 6 uses different interactions with β1 to stabilize alternate conformations in CdiB[Ab] and CdiB[Ec] involving the β1–β16 interface and extracellular loops 1 and 2 (*Figure 3C and D*, *Video 1*).

Structure-based sequence alignment of TpsB transporters allowed us to identify a conserved sequence, the DxxG motif, where 'x' corresponds to polar residues (*Supplementary file 1*). This region is located on β1, interacts with loop 6, and dictates features of the β1–β16 interface on the

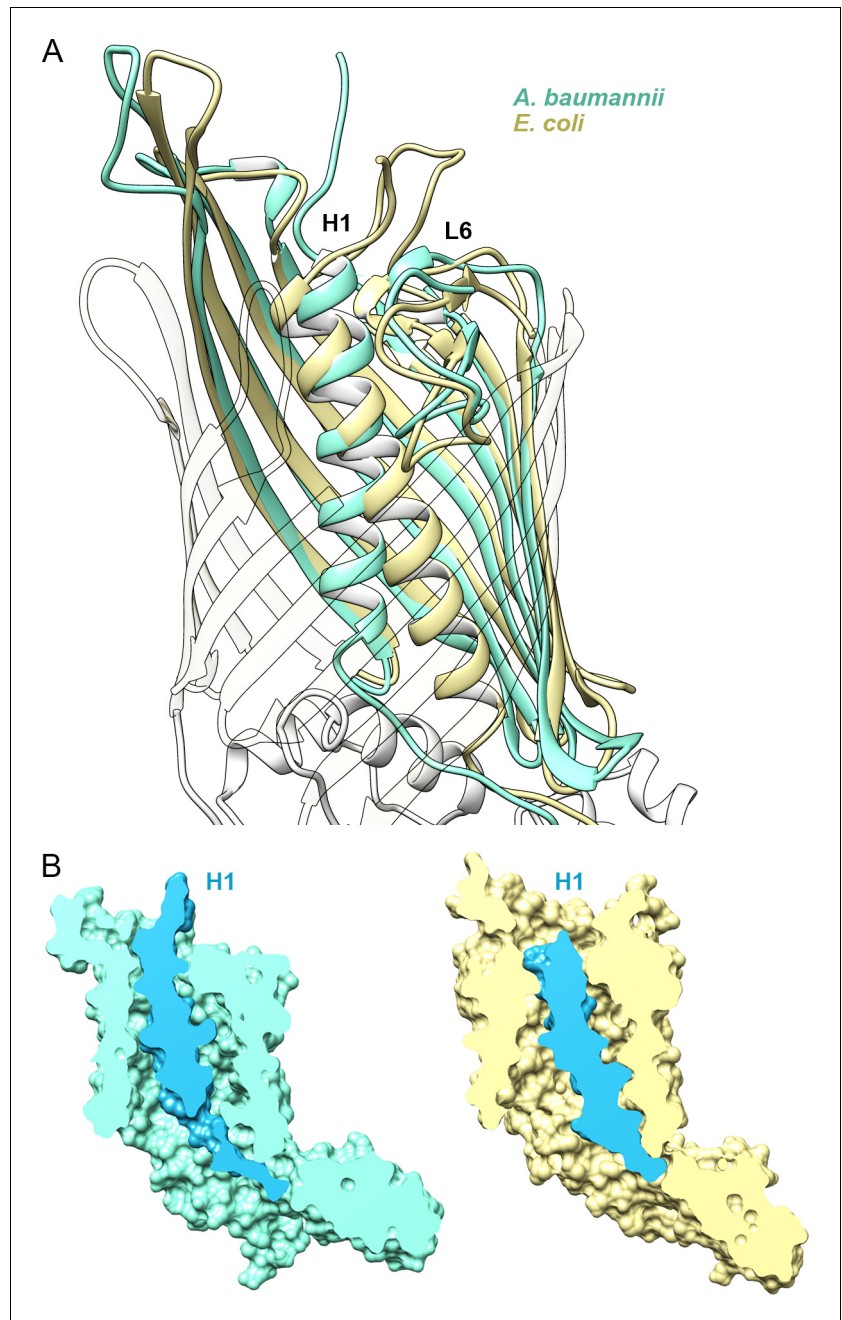

**Figure 2.** Position of Helix H1 in CdiB transporters. (**A**) Membrane view of a superposition of CdiB[Ab] (light teal) and CdiB[Ec] (pale yellow) that illustrates conformational differences of helix H1 inside the β-barrel. To better visualize helix H1, β-strands from the front of the barrel are transparent. (**B**) Molecular surface cross-section of CdiB[Ab] (light teal) and CdiB[Ec] (pale yellow) where helix H1 and linker are shown in blue.

The online version of this article includes the following source data and figure supplement(s) for figure 2:

**Source data 1.** List of interactions involving helix H1 for CdiB[Ab] and CdiB[Ec].
**Figure supplement 1.** RMSD comparison from targeted MD simulations on H1.

extracellular surface of the protein. DxxG can either fully fold as a β-strand or adopt an extended conformation that tilts inward as shown by CdiB[Ab] and CdiB[Ec], respectively (*Figure 3*). In CdiB[Ab], the DxxG motif accounts for four residues on the 12-residue β1 strand, allowing β1 to form an extended β-sheet with β2, β3, and β16 (*Figure 3C*). The sidechains of the first and third residues (D224 and S226) interact with R460 from loop 6 (*Figure 3A*), while the mainchain interacts with

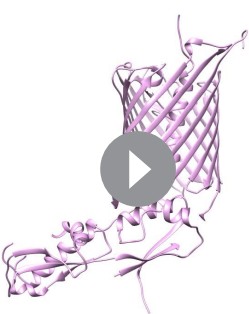

**Video 1.** Morph of CdiB[Ab] and CdiB[Ec] crystal structures. Linear interpolation morph of CdiB[Ab] and CdiB[Ec] crystal structures highlights the conformational differences observed. Helix H1 adopts a different angle and is positioned at a different height in the two structures. β1 twists inward, whereas loop 2 is oriented outward. The POTRA domains are relatively rigid. We note that although the length of the small α-helix in POTRA 1 differs between CdiB[Ab] and CdiB[Ec], the mobility detected in the morph movie is probably an artefact due to the length.

https://elifesciences.org/articles/58100#video1

strand β2. In this structure, the second and fourth residues of the DxxG motif (D225 and G227) form H-bonds with β16. As a result, the β-barrel is in a fully zipped conformation, stabilized by 10 H-bonds between strands β1 and β16 (*Figure 3C*). Extracellular loops 1–2 are short, with loop 2 curved inward and partially blocking the top of the channel (*Figure 3C*). In CdiB[Ec], the DxxG motif is in an extended conformation and initiates loop 1. As a result, strand β1 is shorter, containing only eight residues. The β-sheet formed by β16 and β1-β3 is also shorter (*Figure 3D*). In this structure, the β1–β16 interface is stabilized by only 6 H-bonds, although the sidechain from the second residue (N211) of the DxxG motif contributes to additional interactions with β16 that further stabilize the interface (*Figure 3D*). By increasing the length and flexibility of loop 1, the DxxG motif creates an inward tilting of the loop, allowing it to fold against β2-β3 and interact directly with H1 (*Figure 1—figure supplement 1A and B*). In this position, loop 1 pushes loop 2 away from the lumen of the barrel. In CdiB[Ec], the conserved glycine from the DxxG motif is also involved in a large conformational change by completing the turn made by loop 1 (*Figure 3D*). From our structural analysis, we propose that the role of this region is to increase the flexibility of strand β1, and to facilitate the conformational changes of loop 1 and loop 2.

A comparison of CdiB[Ab] and CdiB[Ec] with TpsB transporter FhaC shows that the structures of FhaC and CdiB[Ec] are virtually identical (*Figure 3—figure supplement 1*), whereas CdiB[Ab] shows an alternate conformation. The high structural similarity observed between FhaC and CdiB transporters suggests that the large body of research pertaining to FhaC is also relevant to CDI secretion.

## Secretion of CdiA by CdiB is specific

To probe the specificity of CDI systems, we generated an *in-vivo* functional assay by co-expressing full-length *cdiB* genes with truncated versions of *cdiA* genes, to produce only the N-terminal domain of the toxin: CdiA-Nt (containing TPS and part of FHA-1) (*Figure 1—figure supplement 3*). After induction, the bacterial pellet was separated from the culture supernatant and production of CdiB and CdiA-Nt were detected by Western blotting. As expected, when CdiB[Ab] and CdiA-Nt[Ab] are co-expressed, CdiB[Ab] is detected in the pellet, and CdiA-Nt[Ab] in the supernatant (*Figure 4A*). As a control, when only CdiB[Ab] is expressed, no CdiA[Ab] secretion is detected, since the protein is not produced. When only CdiA-Nt[Ab] is expressed, neither CdiB[Ab] nor CdiA-Nt[Ab] are detected, since no CdiB[Ab] transporter is available to secrete CdiA-Nt[Ab]. These results show that the secretion of CdiA-Nt is CdiB dependent. We next asked whether CDI systems are specific. No secretion of CdiA occurs when a different CdiB species is used, despite 50% sequence similarity between the two CdiA-Nt constructs. These results show that CDI toxins and transporters are not

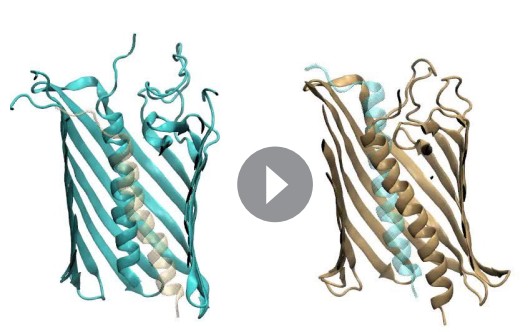

**Video 2.** Targeted Molecular Dynamic simulations of H1 in CdiB[Ab] and CdiB[Ec] models. From the initial state: X-ray structures of CdiB[Ab] (left) or CdiB[Ec] (right), the helix H1 is displaced toward the position of the helix from CdiB[Ec] helix (left) or CdiB[Ab] (right).

https://elifesciences.org/articles/58100#video2

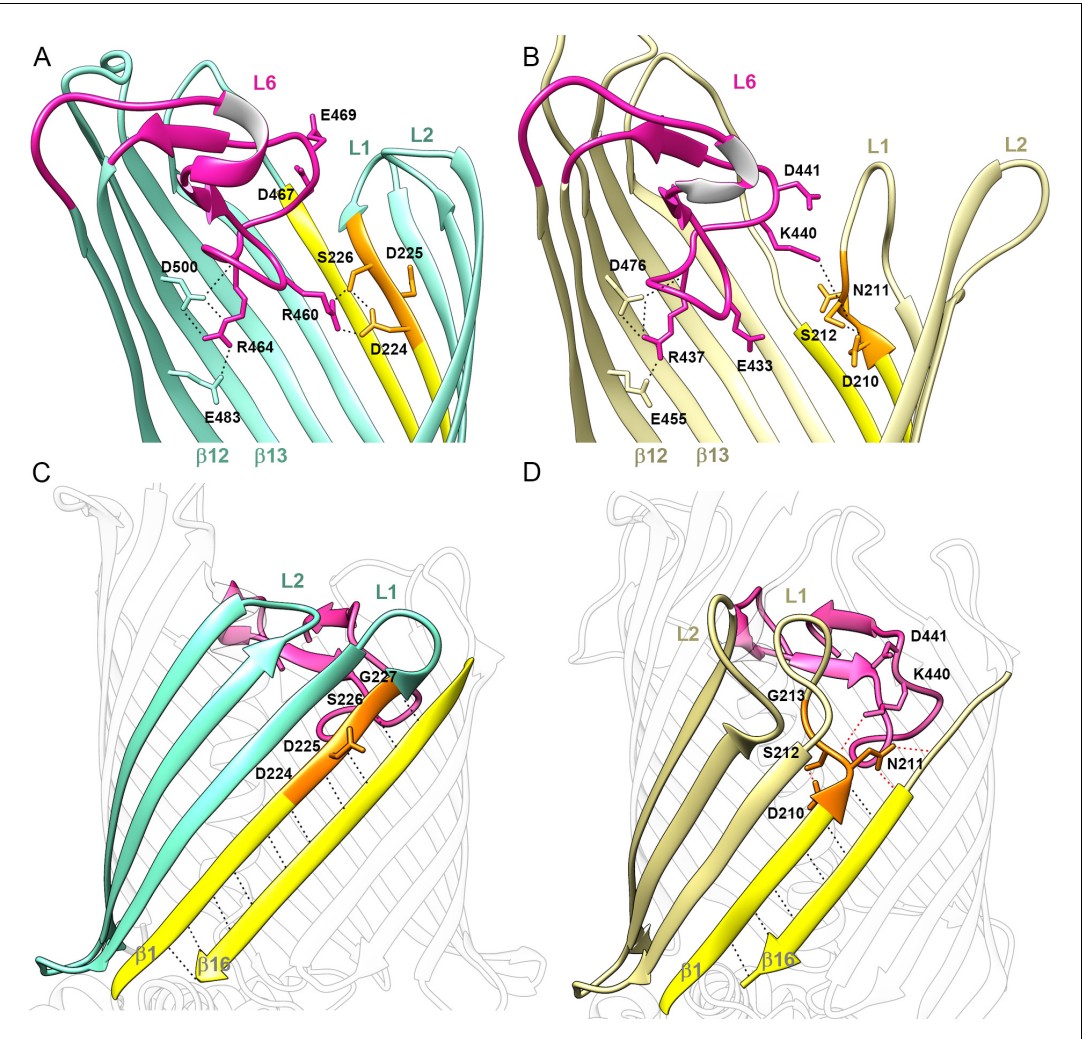

**Figure 3.** Position of Loop 6 and the DxxG motif in CdiB transporters. Zoomed view of loop 6 (L6, magenta) in the lumen of the β-barrel of (**A**) CdiB[Ab] (light teal) and, (**B**) CdiB[Ec] (pale yellow). Sidechains involved in the network of interactions between L6 (VRGF/Y motif), β12 and β13 (F/GxDxG motif) and the DxxG motif (orange) are indicated by black dashed lines. β1 and β16 are colored in yellow, and extracellular loops 1 and 2 are indicated by 'L1' and 'L2', respectively. A conformational difference is shown between CdiB structures: in CdiB[Ab], R460, located before the VRG(F/Y) motif, interacts with the first and third residues of the DxxG motif, while In CdiB[Ec], K440 located after VRG(F/Y), interacts with the second residue of DxxG. Membrane view of the β-barrel of (**C**) CdiB[Ab] and, (**D**) CdiB[Ec] where β2–4 are highlighted in light teal and pale yellow respectively. β1 and β16 are highlighted in yellow, and DxxG motif in orange with important residues numbered. At the β1–β16 interface, mainchain interactions are shown as black dashed lines, and sidechain interactions as red dashed lines. In CdiB[Ec] the two sidechains from L6 pointing toward β1–β16 are shown.

The online version of this article includes the following figure supplement(s) for figure 3:

**Figure supplement 1.** Superposition of CdiB[Ec] and FhaC structures.

interchangeable (*Figure 4A*). As a control, CdiA-Nt[Ec] is detected in the culture supernatant when CdiB[Ec] is expressed. Unfortunately, a small amount of CdiB[Ec] is also detected in the supernatant, indicating that overexpression of CdiB[Ec] can increase bacterial lysis. Therefore, we used only Cdi-B[Ab]/CdiA-Nt[Ab] for subsequent functional analyses.

## Helix H1 and linker influence CdiB stability

To probe the functions of H1 and the linker connecting it to POTRA1, we constructed CdiB variants lacking H1 or lacking both H1 and the linker. The latter construct removes structural elements found in TpsB/CdiB proteins, but not in BamA/TamA proteins, such that only the two POTRA domains and β-barrel are present (*Figure 1—figure supplement 3*). When just the helix is deleted, there is less

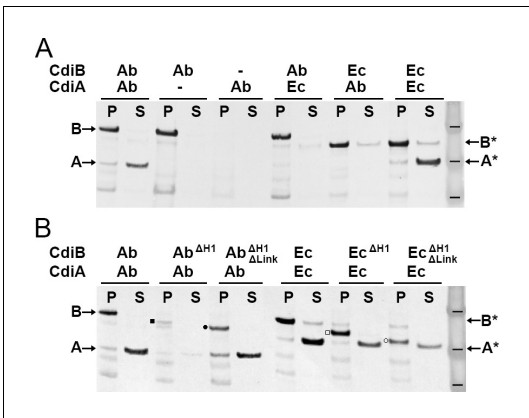

**Figure 4.** Functional analysis of CdiB transporters. Immunoblot analyses of *Escherichia coli* C41 cultures co-expressing CdiB and CdiA-Nt. (**A**) After induction, cell pellets 'P' and culture supernatants 'S' are separated and analyzed to detect the presence of CdiB and CdiA-Nt from *Acinetobacter baumannii* ACICU 'Ab' or *E. coli* EC93 'Ec', respectively. Dash indicates that *cdi* genes are present on the plasmid but not induced. CdiA-Nt$^{Ab\ 74-472}$ and CdiA-Nt$^{Ec\ 29-416}$ are only detected in the supernatant when their respective CdiB transporters are present. (**B**) CdiB variants where helix H1 'ΔH1' and Linker 'ΔLink' are genetically deleted and co-expressed with their cognate CdiA-Nt substrates from CDI systems of *A. baumannii* 'Ab' and *E. coli* 'Ec'. CdiB and CdiA-Nt wildtype proteins detected in immunoblot are indicated by B and A for *A. baumannii* ACICU and B* and A* for *E. coli*. Detection of CdiB$^{ΔH1}$ and CdiB$^{ΔH1ΔLink}$ variants are indicated respectively, by a black square and a black circle for *A. baumannii*; and a white square and a white circle for *E. coli*. Protein ladder bands indicate 62, 49, 38 kDa respectively.

CdiB in the pellet fraction, especially for CdiB$^{Ab}$, and correspondingly, a smaller amount of CdiA-Nt is secreted (*Figure 4B*). This suggests that H1 is important for folding, membrane insertion, and/or stability of CdiB. However, when both the helix and linker are deleted, CdiB is present and secretion of CdiA-Nt occurs (*Figure 4B*). These results show that the helix and linker are not essential for CdiA-Nt secretion but must be important for CdiB stability. In comparison, the linker was found to be essential for substrate secretion in FhaC and helps to stabilize the POTRA domains (*Delattre et al., 2011*; *Jacob-Dubuisson et al., 2009*). However, the precise function of the linker and why its presence alone drastically affects CdiB$^{Ab}$, are unclear. One obvious function of H1 is to plug the β-barrel lumen when substrate is absent, preventing entry/exit of unwanted molecules (*Clantin et al., 2007*).

## Flexibility of the β1–β16 interface is essential for secretion

In CdiB$^{Ab}$, the DxxG motif increases the length of the first β-strand to 12 residues, rigidifying the β1–β16 interface, but this region appears more flexible in CdiB$^{Ec}$ and FhaC structures (*Figure 3*, *Figure 5C*, *Video 1*, *Figure 3—figure supplement 1*). To understand whether the flexibility of DxxG is important for activity, we engineered paired cysteine variants between β1 and β16 to stabilize the conformation of CdiB$^{Ab}$. The CdiB$^{Ab}$ cysteine variants were then tested for their ability to secrete CdiA-Nt$^{Ab}$ in the absence or presence of the reducing agent TCEP. Disulfide bond formation was analyzed by Western blot (*Figure 5*). Clear disulfide bond formation was observed for all mutants positioned in the middle of the β-sheet. When β1 and β16 are cross-linked, substrate secretion is greatly impaired, however reduction of the disulfide rescues secretion (*Figure 5A*). To confirm that crosslinking β1–β16 does not impair CdiB$^{Ab}$ biogenesis, we monitored expression of CdiB D224C/S555C and secretion of CdiA-Nt over time (*Figure 5B*). CdiB$^{Ab}$ was detected in the pellet from 20 to 100 min after induction, in the presence or absence of TCEP. In comparison, the secretion of CdiA-Nt is greatly increased only when the β1–β16 disulfide is reduced with TCEP. To confirm that the disulfide mutants are correctly targeted to the outer membrane, we isolated and solubilized membranes from the D225C/F554C mutant. The oxidized form was detected in membranes and could be solubilized with detergent (*Figure 5—figure supplement 1A*). As a control, we confirmed disulfide formation in a strain lacking the periplasmic oxidoreductase, DsbA (*Figure 5—figure supplement 1B*). Altogether, these results show that secretion of CdiA is inhibited when strands β1 and β16 are tethered, and correspondingly, flexibility of β1 and the DxxG motif facilitate secretion.

## Loop 2 and the DxxG motif influence secretion

Based on conformational differences in the CdiB crystal structures, we probed residues that might be important for CdiA secretion. We monitored CdiA secretion over 130 min (*Figure 6A*, left). We controlled the system so that production does not increase cell lysis, and we monitored the expression levels of CdiB in the bacterial pellet (*Figure 6A*, right; *Figure 6—figure supplement 1A*). For

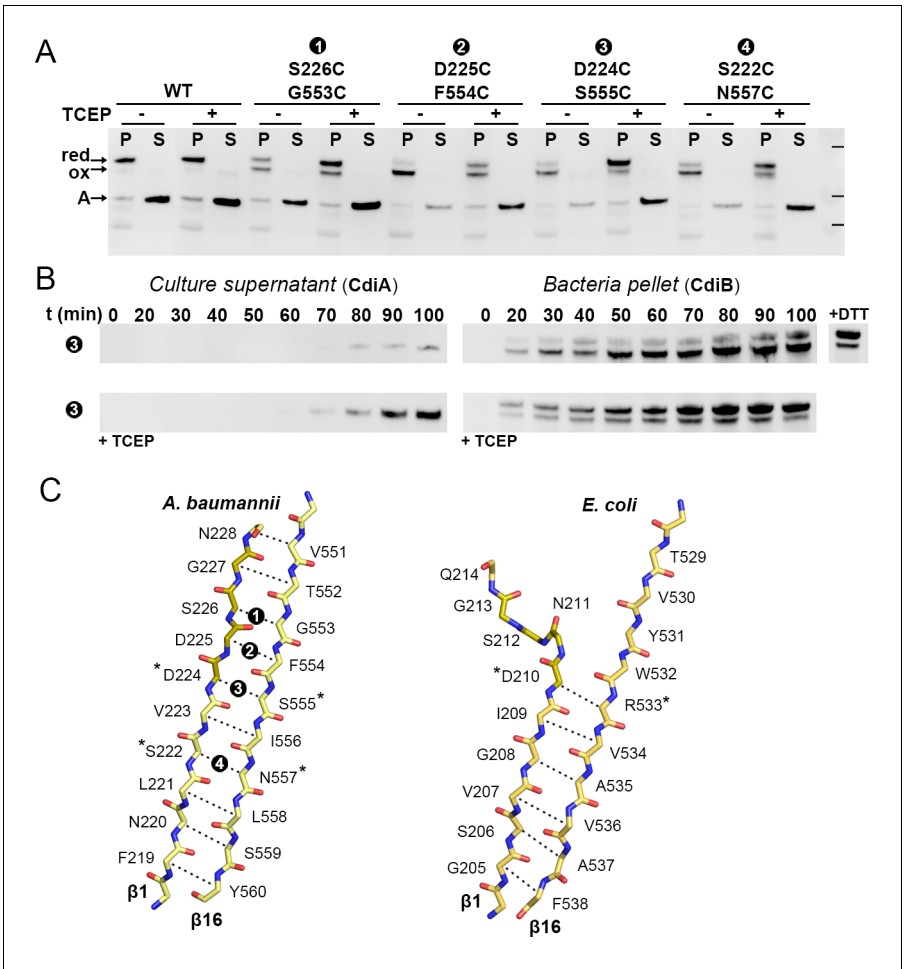

**Figure 5.** β1–β16 disulfide crosslinking in CdiB^Ab. (A) CdiB^Ab cysteine variants are co-expressed with CdiA-Nt substrate from *Acinetobacter baumannii* in presence + or absence - of reducing agent 'TCEP' in the culture. After induction, pellet 'P' and supernatant 'S' are separated and analyzed by western blot to detect the presence of CdiA-Nt 'A' and, CdiB without cross-linking 'red', or CdiB with β1–β16 crosslinked 'ox'. Protein ladder bands indicate 70, 50, 40 kDa, respectively. (B) After induction (t0) the secretion of CdiA-Nt and the production of CdiB^Ab double cysteine D224C-S555C are monitored at 10 min intervals. Samples from culture supernatant and bacterial pellet are separated, analyzed by western blot, and the bands corresponding to CdiA-Nt (left) and CdiB^Ab (right) displayed. For comparison, reducing agent was added to the pellet fraction t100 ('+DTT') or incubated with the culture at t0 ('+TCEP'). (C) Mainchain representation of β1–β16 of CdiB from *A. baumannii* (left) and *E. coli* (right) colored in yellow, or orange for the DxxG motif. Adjacent sidechains oriented in the same direction are indicated by dashed lines, where * indicate sidechains involved in H-bonding in the crystal structure. Black circles with white numbers indicate engineered disulfides.

The online version of this article includes the following figure supplement(s) for figure 5:

**Figure supplement 1.** Expression of CdiB^Ab double cysteine variant.

all mutants, CdiB was detected after 20 min and levels increased over time. In contrast, levels of secreted wildtype CdiA-Nt are first detected after 60 min. Removing H1 and the linker (WT^ΔH1ΔLink) slightly improves secretion, with the toxin first detected after 50 min. This result reconfirms that H1 is not essential for TpsB function (*Méli et al., 2006*; *Figure 4B*).

Loop 2 adopts very different conformations in our two CdiB structures. Whereas deletion of this loop prevents expression of CdiB, the 4-residue substitution of DDFH to GGAG is tolerated and does not increase cell lysis (*Figure 6—figure supplement 1B*). Removal of the sidechains can prevent loop 2 interaction with the rest of the protein while increasing loop 2 flexibility. The GGAG mutant results in measurable impact on the secretion, where CdiA is detected after just 30 min and

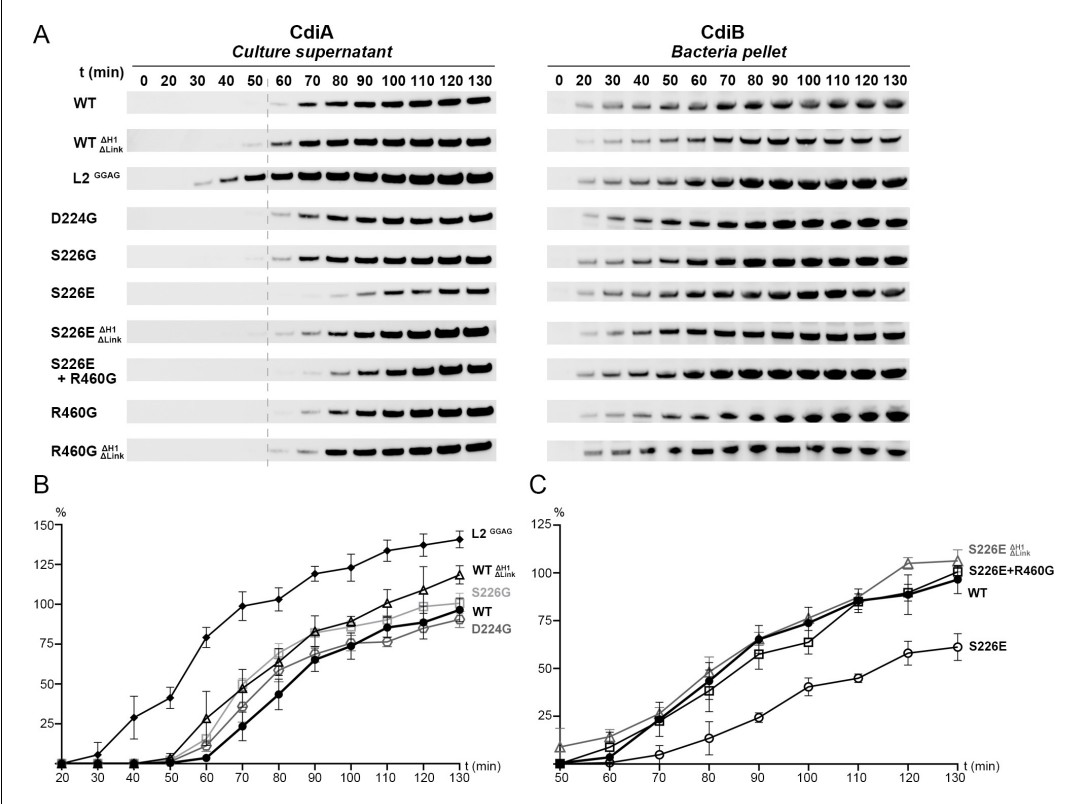

**Figure 6.** Analysis of CdiB$^{Ab}$ secretion activity. (**A**) Secretion of CdiA-Nt$^{Ab\ 74-472}$ and expression of CdiB$^{Ab}$ are monitored after induction (**t0**) up to 130 mins. Samples from culture supernatant and bacterial pellet were separated by centrifugation, analyzed by western blot, and the bands corresponding to CdiA-Nt (left) and CdiB$^{Ab}$ variants (right) displayed. A gray dashed line at 60 min after induction indicates the reference point for the W.T where enough CdiA-Nt is present in the culture supernatant for immunoblot detection. (**B**), (**C**) Secretion activity was assessed by the levels of CdiA-Nt$^{Ab\ 74-472}$ detected in culture supernatants normalized to the wildtype at t130 min (100%), repeated three times independently where error bars represent the standard error of the mean. (**B**), Secretion activity comparison from 20 to 130 min for CdiB$^{Ab}$ wildtype (WT, •), CdiB where H1 and Linker are genetically deleted (WT$^{\Delta H1 \Delta Linker}$, Δ), Loop 2 substitutions GGAG (L2$^{GGAG}$, ♦) and single substitution variants D224G (○, gray), S226G (□, light gray). (**C**), Secretion activity comparison from 50 to 130 min between CdiB$^{Ab}$ wildtype (WT, •), single substitution S226E (○), S226E without H1 and Linker (S226E$^{\Delta H1 \Delta Linker}$, Δ gray), and double substitution S226E+R460G (□).

The online version of this article includes the following figure supplement(s) for figure 6:

**Figure supplement 1.** Western blot analysis of GroEL and MBP in wild type and L2$^{GGAG}$ cultures.

continues to increase over time. This result suggests that increasing loop 2 flexibility enhances secretion activity and may promote the active conformation of the β-barrel. As a comparison, when loop 2 adopts an inward conformation in the CdiB$^{Ab}$ X-ray structure, the sidechains point in the direction of H1 and loop 6, and the β-barrel lumen is partially capped (**Figure 1**, **Figure 3**, and **Video 1**).

Our structural and functional analysis reveal that β1 can be fully folded as a β-strand stabilized by β16 or adopt a more flexible extended secondary structure and be folded inward as part of loop 1. Our hypothesis is that the conserved DxxG motif can facilitate a conformational change that promotes the active conformation. To explore the role of the interaction between DxxG and the conserved loop 6 inside the β-barrel, we made point mutants in the D224-S226-R460 interaction network. Mutations to glycine do not lower the rate of CdiA secretion, indicating no essential role for these residues in producing an active CdiB conformation and/or substrate interaction (**Figure 6A and B**). However, the CdiB$^{Ab}$ structure predicts that mutation of S226 to glutamate would allow formation of a salt bridge with R460, further stabilizing the DxxG-L6 network. In fact, S226E delays secretion and results in lower amounts of CdiA secreted over time. Combining S226E with R460G or with the H1-linker deletion (S226E$^{\Delta H1 \Delta Link}$) improves secretion and restores CdiA levels to near

wildtype (*Figure 6A and C*). These results show that interactions between DxxG and loop 6 affect CdiA secretion and that flexibility in this region is essential.

## Link between DxxG conformation and position of H1 helix

The main conformational differences observed between CdiB[Ab] and CdiB[Ec] are at the β1–β16 interface at the DxxG motif, and inside the barrel with H1. Based on structural analysis of CdiB[Ec], the rearrangement of loop 1 created by DxxG forms a network of interactions between loop 1, H1, and loop 6 (Q214-R10-K440-N211; *Figure 1—figure supplement 1*). Interestingly, in the FhaC structure, there is a similar network of interactions between β1, H1, and L6, where R17 from H1 interacts with the conserved aspartate from DxxG (*Figure 3—figure supplement 1*). In both CdiB[Ec] and FhaC structures, the inward tilting of DxxG in β1 promotes the rearrangement of loop 1 and loop 2 while presenting a new interaction surface for H1. However, due to the weak sequence conservation of H1 in TpsB transporters (*Supplementary file 1*), it is difficult to predict whether particular residues either stabilize the helix in the barrel pore or induce its exit in the active conformation. Since the position of H1 in our CdiB structures vary, we wanted to understand whether β1–β16 and DxxG remain flexible. Using molecular dynamics, we ran three 500-ns equilibrium simulations of each CdiB structure in a species-specific outer membrane (*Video 3*, *Figure 7—figure supplement 1*; *Phillips et al., 2005*). Although the position of H1 remains stable in the β-barrel lumen, we observed rupture of several H-bonds between β1 and β16 in the CdiB[Ab] simulations. Part of the DxxG motif can convert from a β-strand to a loop, whereas the β1–β16 interface can fluctuate from a short to a long β-sheet affecting the size of the β16-β1-β2-β3 sheet. During the simulations we observe that water molecules penetrate into the membrane slightly to keep the DxxG motif solvated (*Figure 7—figure supplement 1C*). This environment may facilitate the interconversion of secondary structure. In comparison, during CdiB[Ec] simulations, the DxxG motif tilts inward, toward the lumen, and the CdiB[Ec] conformation is much more stable, with no H-bond disruption detected between β1 and β16. This result shows that DxxG can exist in two different conformational states: fully folded as a β-strand or in a more flexible, extended conformation. The equilibrium simulations demonstrate that CdiB[Ab] interconverts between the two conformations, while CdiB[Ec] stabilizes only the unfolded conformation. However, the importance of these structural differences to CdiA secretion remains to be determined.

## Extraction of helix H1 from β-barrel lumen

To secrete substrates across the outer membrane, TpsB/CdiB transporters must cycle through multiple conformational states, from the resting conformation with H1 inserted into the β-barrel lumen, to the active form when H1 resides in the periplasm (*Figure 1—figure supplement 3*; *Baud et al., 2014*; *Guérin et al., 2014*). To understand these conformational changes, we induced the exit of H1 by <u>s</u>teered <u>m</u>olecular <u>d</u>ynamics (SMD) to measure the force needed to extract it from the pore and observe its exit path (*Video 4*, *Video 5*; *Sotomayor and Schulten, 2007*). H1 was pulled in the direction of the periplasm at a constant speed of 0.29 Å/ns over 150–200 ns. In these experiments, we wanted to mimic a hypothetical periplasmic force that could cause the helix to exit the pore as might be expected when substrate is present and/or when large conformational changes in the linker and POTRA domains occur. During the simulations, H1 is free to move or rotate as force is applied to the center of mass of the helix. We

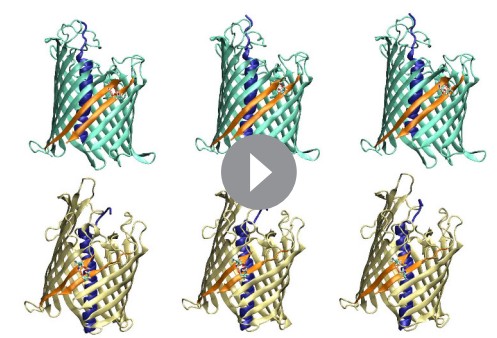

**Video 3.** Equilibrium simulation of CdiB[Ab] and CdiB[Ec]. 500 ns equilibrium simulations of three CdiB[Ab] (light teal) and three CdiB[Ec] (pale yellow) molecules. Full length CdiB proteins are inserted into their respective species-specific outer membranes (shown in *Figure 7—figure supplement 1*). Strands β1 and β16 are highlighted in yellow, and helix H1 in blue. The H-bonds between G227-T552 and I229-V534 are indicated for CdiB[Ab] and CdiB[Ec], respectively. The POTRA domains and linker were present for the simulations but not shown in the final movies.
https://elifesciences.org/articles/58100#video3

tracked the position of H1 and measured the force required. We also used the SMD trajectories to seed potential of mean force (PMF) calculations to determine the free energy required to extract H1 from CdiB$^{Ab}$ and CdiB$^{Ec}$ β-barrels (*Figure 7*, *Figure 7—figure supplement 2A*; *Sugita et al., 2000*). Since the exit pathway of H1 is not known, we ran four independent simulations (two per species) to assess the role of sampling during SMD simulations and PMF calculations. In the case of CdiB$^{Ab}$, both independently determined PMF profiles show a continuous rise in energy as the helix is extracted from the barrel. With large energetic barriers measured at 90kcal/mol and 32kcal/mol, respectively, the exit of H1 appears to be energetically expensive (*Figure 7—figure supplement 2A*). Similarly, the first PMF determined for H1 extraction from CdiB$^{Ec}$ suggests 35 kcal/mol is required to extract H1. However, the second run for CdiB$^{Ec}$ found a significantly lower energy path on a similar time scale as the other PMF calculations (35–60 ns/window, or 1.7–2.6 µs in total) (*Figure 7*). Therefore, we extended this PMF calculation to 235 ns/window (10.1 µs in total). From this PMF, we discovered a new minimum, one even lower in energy than the crystal structure minimum, at a point of intermediate extraction. In this PMF, the second minimum is separated from the crystal-structure conformation by 7.5 kcal/mol and a barrier of 13 kcal/mol; the fully extracted state is 7.5 kcal/mol higher than the second minimum and almost identical to the crystal-structure conformation. The existence of an intermediate state during helix extraction from the barrel is supported by earlier experimental data where Pulsed-electron double-resonance (PELDOR) spectroscopy revealed distinct peaks in the distance distribution on a related two-partner-secretion transporter, FhaC, even in the absence of any substrate (*Guérin et al., 2014*). We note that for computational efficiency, the POTRA domains were not present in the PMF calculations; along with the substrate, they may shift the relative energies of the fully embedded, partially embedded, and fully extracted H1 conformations. Also, the different PMFs obtained for different starting conditions illustrates the systematic uncertainty present in these calculations when run for moderate lengths (1–2 µs in total), complicated further by the unknown end-state structure.

To better understand the exit pathway and investigate any conformational differences between CdiB$^{Ab}$ and CdiB$^{Ec}$, we ran several additional SMD simulations. Using the X-ray structures as an initial model, we pulled the H1 helix to the periplasm and quantified the force as a function of the position during extraction for three replicas (*Figure 7—figure supplement 2B*). The force plots for CdiB$^{Ec}$ and CdiB$^{Ab}$ display distinct features, while on average CdiB$^{Ec}$ requires less work to extract H1, especially from −25 to −35 Å. Upon further examination, we observed a strong electrostatic interaction between H1 and the DxxG motif or loop 2 in CdiB$^{Ab}$ that require a large force to disrupt (*Figure 7—figure supplement 3*). This ion-bridged interaction is not observed for CdiB$^{Ec}$, probably because loop 2 is oriented outward and does not interact with H1. Based on the CdiB$^{Ab}$ functional data (*Figure 5*, *Figure 6*) we built and simulated two CdiB$^{Ab}$ mutants. The force plot of CdiB$^{Ab}$ in which loop 2 is deleted shows that less force is required to extract H1 on average. In comparison, the force plots from the disulfide-bonded β1–β16 mutant display features similar to the wildtype CdiB$^{Ab}$, with multiple peaks observed in the −10 to −35 Å range (*Figure 7—figure supplement 2B*). These results show that by stabilizing H1 in the barrel lumen through electrostatic interactions, some structural elements (such as loop 2 and the DxxG motif) influence the energetic barrier needed to open the β-barrel pore.

Altogether our simulations indicate that the exit of H1 must follow a pathway inside the β-barrel lumen to be correctly extracted. As an alternative, conformational changes from the β-barrel itself (rearrangement of extracellular loops, flexibility of DxxG motif) decrease the energy required to eject H1 while probably preparing the active conformation.

## Discussion

By releasing large exoproteins at the surface of the cell, type Vb secretion systems play an essential role in pathogenesis and survival of Gram-negative bacteria. Genome databases have reported hundreds of TpsA proteins that come in different sizes and domain organizations, where CdiA toxins represent a special subgroup. An increasing number of studies have reported multifunctional roles for CdiA proteins, apparently independent of toxin activity. CdiA promotes adhesion on epithelial cells in *A. baumannii,* plays a major role in intracellular survival and intracellular escape in *Neisseria meningitidis*, increases the virulence of *Pseudomonas aeruginosa* in infection models, and controls biofilm establishment in human pathogens (*Melvin et al., 2017*; *Mercy et al., 2016*; *Pérez et al.,*

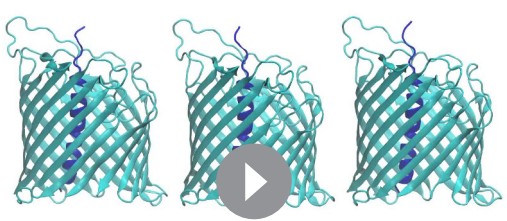

**Video 4.** Extraction of CdiB[Ab] helix H1 by Steered Molecular Dynamics. SMD simulations of CdiB[Ab] (light teal) where helix H1 is pulled toward the direction of the periplasm at a constant speed (0.29 Å/ns). https://elifesciences.org/articles/58100#video4

2016; *Roussin et al., 2019*; *Talà et al., 2008*). Both CDI systems investigated in our study have been shown to be constitutively active, and a secretome analysis of *Acinetobacter spp.* revealed that CdiA is one of the most abundantly secreted proteins (*Aoki et al., 2005*; *Harding et al., 2017*).

During the secretion cycle, the TpsB β-barrel must adopt multiple conformational states favoring ejection of the N-terminal helix H1, entry and movement of substrate into the β-barrel lumen, folding of substrate in the extracellular space and then re-entry of H1 into the lumen. All steps happen in an environment where no hydrolysable energy or electrochemical gradient sources are available to power conformational changes. The limited space inside the β-barrel lumen and the multiple tertiary interactions with H1 increase the energetic barrier needed to eject the internal α-helix. As a result, additional conformational changes are likely required to obtain the active conformation, such as potential rearrangement of the β-barrel itself. These changes may be induced by the presence of the substrate CdiA, the binding of which to the POTRA domains could provide the additional 7.5 kcal/mol necessary to fully extract the helix (*Figure 7*).

The flexibility of the β1–β16 interface mediated by the DxxG motif plays an essential role in the transport mechanism, where conformational changes in loop 1 display different interacting surfaces for H1 and possibly for the toxin substrate. At the β1–β16 interface, sidechains from the conserved loop 6 interact with the DxxG motif, helping to stabilize different conformations. Our structures also reveal two different conformations for loop 2 that affect CdiA secretion and H1 ejection. Similar results have been observed for FhaC, where deletion of this loop does not affect the active conformation, whereas insertion of several residues reduces the channel conductance and prevents substrate secretion (*Baud et al., 2014*; *Méli et al., 2006*). These results suggest a common and conserved role of loop 2 in the TpsB transporter family, where the inward state can stabilize the resting conformation, and the outward state facilitate the active conformation.

H1 and the DxxG motif are both present in CdiB/TpsB transporters, but absent in BamA/TamA proteins. We hypothesize that they contribute to unidirectional secretion where the lumen of the β-barrel is accessible or inaccessible at different stages of the secretion cycle. The position of loop 6 and interactions with β13 are conserved in all Omp85 proteins, suggesting a common function. In addition to possibly interacting with the substrate upon entering the β-barrel lumen, our results suggest that loop 6 can also stabilize different conformational states of the β-barrel at the β1–β16 interface.

## Materials and methods

### Cloning and purification of CdiB proteins

cdiB[Ab] from *A. baumannii* strain ACICU (locus tag ACICU_01912, protein id WP_000956371) and cdiB[Ec] from *E. coli* strain EC93 (locus DQ100454, protein id AAZ57197.1) were codon-optimized, synthesized (Genscript) and cloned using ligation- independent cloning into pET9

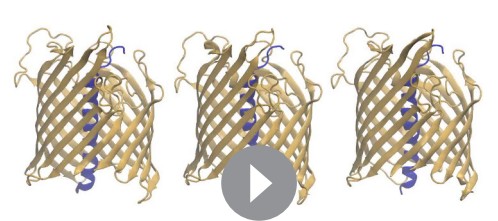

**Video 5.** Extraction of CdiB[Ec] helix H1 by Steered Molecular Dynamics. SMD simulations of CdiB[Ec] (pale yellow) where helix H1 is pulled toward the direction of the periplasm at a constant speed (0.29 Å/ns). https://elifesciences.org/articles/58100#video5

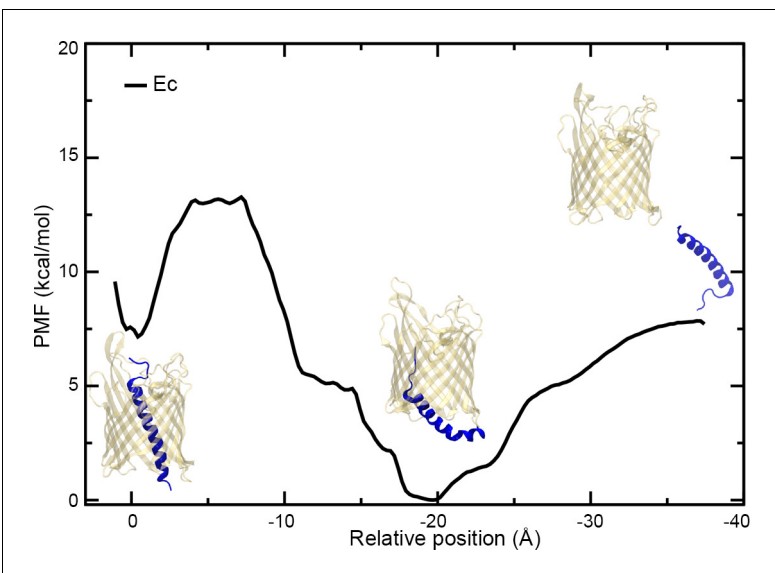

**Figure 7.** PMF of the extraction of H1 from the CdiB[Ec] β-barrel. Plot of the potential of mean force (PMF) measured as a function of the distance between the centers-of-mass of the α-helix H1 and β-barrel of CdiB[Ec]. Three images of CdiB[Ec] have been added to the plot representing different states, from the left to right: resting conformation, intermediate extraction, and H1 out of the pore.

The online version of this article includes the following figure supplement(s) for figure 7:

**Figure supplement 1.** Initial states of the CdiB[Ab] and CdiB[Ec] simulations.

**Figure supplement 2.** Extraction of H1 from the CdiB β-barrel.

**Figure supplement 3.** Examples of ion-bridged interactions in CdiB[Ab].

vector, a derivative of pET20b (EMD Millipore). Signal sequence positions 1–23 for CdiB[Ab] and 1–52 for CdiB[Ec] were replaced by the pelB signal sequence followed by a 10-Histidine tag and a TEV site (ENLYFQSM) added to the N-terminus of mature proteins. Expression was performed in BL21(DE3) cells in 12 liters of TB media supplemented with 25 µg/mL kanamycin during 3 days at 20°C without induction (leaky expression of pET9 vector). Cells were collected by centrifugation (7500 g for 15 min), and resuspended in lysis buffer (50 mM Tris-HCl, pH7.4, 200 mM NaCl, 1 mM MgCl$_2$, 10 mg/mL DNase I, and 100 mg/mL 4-(2-aminoethyl)benzenesulphonyl fluoride (AEBSF)). Cells were broken by three passages through an Emulsiflex C3 (Avestin) homogenizer at 4°C, and unlysed cells removed by centrifugation (7500 g for 15 min). The membrane fraction was harvested by ultracentrifugation (160,000 g for 60 min), and the pellet was resuspended in 50 mM Tris-HCl, pH7.4, 200 mM NaCl, 20 mM imidazole and solubilized by constant stirring in 5% Elugent (EMD Millipore) for 16 hr at 4°C. Solubilized membranes were harvested by a second ultracentrifugation step (220,000 g for 60 min) and the supernatant containing CdiB proteins was applied to a 15 mL Ni-NTA column (Qiagen) and eluted with 50 mM Tris-HCl, pH7.4, 200 mM NaCl, 0.8% Elugent and 250 mM imidazole. To remove the N-terminal 10-His Tag, peak fractions were pooled and incubated with 2 mg of TEV protease, in the presence of 2 mM DTT and 1 mM EDTA at 4°C under gentle agitation for 12 hr. The mixture was diluted into 50 mM Tris-HCl, pH8, 0.8% elugent and applied to an anion exchange chromatography column (Q sepharose GE Healthcare) for detergent exchange, eluted using a NaCl gradient into 50 mM Tris-HCl, pH8, and 1% C$_8$E$_4$ (Anatrace). To remove uncleaved protein from the TEV digestion, the peak fractions from ion-exchange were applied to a second Ni-NTA purification on gravity column using 2 mL of resin. The flow through containing TEV-digested CdiB proteins was concentrated to 4 mL and applied to a HiLoad 16/600 Superdex 200 size exclusion column (GE Healthcare) using 25 mM NaPi pH6.6, 100 mM NaCl, 1% C$_8$E$_4$. For Selenomethionine-substituted CdiB[Ab] proteins, expression was performed in B834 *E. coli* cells (Novagen). Cultures were started in 12 L of TB media, then when OD$_{600}$ reached 0.8, cells were harvested and washed two times in SelenoMet minimal media (Molecular Dimensions) supplemented with L-methionine at 60 mg/L. The final round was resuspended in 1 liter of SelenoMet media to inoculate 12 liters of SelenoMet

supplemented with L-methionine at 60 mg.L$^{-1}$ and 50 µg.mL$^{-1}$ kanamycin. When OD$_{600}$ reached 0.7, SeMet derivatized CdiB$^{Ab}$ proteins were induced by addition of 1 mM Isopropyl-thio β-D-1-thiogalactopyranoside (IPTG) and grown 16 hr at 30°C. The final OD$_{600}$ was ~2.5, cells were harvested by centrifugation and CdiB proteins purified as described above. The incorporation of selenium into CdiB$^{Ab}$ proteins was analyzed by mass spectrometry (data not shown; Taplin – Harvard).

## Crystallization and data collection

For crystallization, samples were concentrated to ~10 mg/mL and sparse matrix screening was performed using a TTP Labtech Mosquito crystallization robot using hanging drop vapor diffusion with plates incubated at 21°C. The best native crystals for CdiB$^{Ab}$ were grown from 100 mM Tris-HCl pH 8.4, 200 mM lithium sulfate, 10% PEG400, and 23% ethylene glycol. Selenomethionine-substituted crystals of CdiB$^{Ab}$ were crystallized using similar conditions to native: 100 mM Tris-HCl pH 8, 200 mM lithium sulfate, 11% PEG400, and 23% ethylene glycol. The best crystals for CdiB$^{Ec}$ were grown from Morpheus II condition C10 (Molecular Dimensions): 100 mM Gly-Gly, AMPD pH8.5, 4 mM Alkalis (1 mM Rubidium chloride, 1 mM Strontium acetate, 1 mM Cesium acetate, 1 mM Barium acetate), 12.5% PEG4000% and 20% 1,2,6-Hexanetriol. Crystals were collected directly from the crystallization drops and native data were collected at SER-CAT (ID22) and the GM/CA-CAT (ID23-D) beamlines of the Advanced Photon Source of the Argonne National Laboratory. Data collection for selenium-single-wavelength anomalous dispersion (Se-SAD) phasing of CdiB$^{Ab}$ was performed at the BL12-2 beamline of the Stanford Synchrotron Radiation Lightsource from the SLAC National Accelerator Laboratory, during the Rapidata practical course. A summary of the data collection statistics can be found in *Table 1*.

## Structure determination

Molecular replacement on CdiB$^{Ab}$ native data using the FhaC structures (PDB 4QKY and 3NJT) was unsuccessful. We phased the CdiB$^{Ab}$ structure by collecting one data set on selenomethionine substituted CdiB$^{Ab}$ crystal at the wavelength 0.979 Å. The data were processed in space group P1 to a final resolution of 2.6 Å and selenium sites located using SHELX (*Sheldrick, 2010*). A phase-extended density-modified electron density map was produced with AutoSol (PHENIX) (*Adams et al., 2010*) and used for iterative model building (COOT [*Emsley and Cowtan, 2004*]) and refinement (PHENIX). This model was then used as a search model to solve the selenomethionine derivative CdiB$^{Ab}$ and native CdiB$^{Ec}$ structures by molecular replacement using Phaser-MR (*Adams et al., 2010*). The CdiB$^{Ab}$ structure was refined in space group P1 to 2.4 Å resolution with R/R$_{free}$ values of 0.20/0.25 and CdiB$^{Ec}$ in space group P2$_1$2$_1$2 to 2.6 Å resolution with R/R$_{free}$ values of 0.24/0.26. Figures were made with UCSF Chimera (*Pettersen et al., 2004*).

Coordinates and structure factors for the CdiB$^{Ab}$ and CdiB$^{Ec}$ structures have been deposited in the Protein Data Bank (PDB accession codes 6WIL and 6WIM).

## Sequence alignments

Starting after the predicted signal sequence, sequence alignments included representative Type Vb transporters where eight are involved in the CDI mechanism (6 CdiB: WP_000956371, AAZ57197.1, WP_046042815, ACI07001.1, WP_002210394.1, NP_273542, WP_126867950. 2 BcpB: WP_011402463, WP_011851264) and nine representative TpsB transporters, not involved in the CDI mechanism (WP_010930614, VDH07240, BAA21096, WP_136264517, AAA50322, AAA87060.1, AAX13508.1, WP_011191836, WP_010895677). Alignments were performed with T-coffee (*Notredame et al., 2000*) and edited with Jalview (*Waterhouse et al., 2009*) to take into consideration the secondary structure from available structural data. The final result was presented with ESPript (*Robert and Gouet, 2014*). Sequence identity percentage was calculated using Blastp suite (NCBI).

## Secretion assays

To test secretion activity, two plasmids were used to co-produce CdiB and CdiA-NT proteins in *E. coli* C41 (DE3) cells. cdiB genes were cloned into the pBAD plasmid under the control of the arabinose promoter, where the signal sequences (1–23 for CdiB$^{Ab}$ and 1–52 for CdiB$^{Ec}$) were replaced by

a pelB signal sequence, and 6-Histidine tags inserted into extracellular loop7 (located between residues 510–511 for CdiA$^{Ab}$, and between residues 486–487 for CdiA$^{Ac}$). The 5' region of cdiA genes from the related CDI operons was codon-optimized and synthesized (Genscript) to produce region 74–472 for CdiA$^{Ab}$ (WP_001039234.1) and region 29–460 for CdiA$^{Ec}$ (AAZ57198.1) containing the TPS domain and part of the FHA-1 domain. cdiA-NT genes were cloned into a pCDF plasmid under the control of a T7 promoter with lac operator, and the native signal sequence (1–73 and 1–28) was replaced by the pelB signal sequence, and a 6-Histidine tag added at the C-termini of the proteins. Cloning and CdiB deletion variants (denoted Δ in the text) were engineered using the Gibson assembly method (NEB) and amino acids substituted using Q5 Site-Directed Mutagenesis (NEB). α-helix H1 deletion constructs were built by removing the first 26 (IEDVSLPSQVLQDQRLKELNQQLQDQ) and 29 (AMLSPGDRSAIQQQQQQLLDENQRQRDAL) N-terminal residues from CdiB$^{Ab}$ and CdiB$^{Ec}$, respectively. Constructs with α-helix H1 and linker deleted start at the first conserved cysteine indicated in *Supplementary file 1* (end of the linker). For secretion assay *E. coli* C41(DE3) cells were co-transformed using pBAD-cdiB and pCDF-cdiA-NT constructs and selected with 25 μg/mL kanamycin and 20 μg/mL streptomycin on LB (LB K/S). Cultures for secretion assays were incubated by shaking at 37°C 20 mL culture LB K/S in 125 mL flasks. When cultures reached $OD_{600}$ = 0.8, 0.1% arabinose and 400 μM IPTG were added for exactly 2h30 min and standardized at the end of the induction period using the final $OD_{600}$, 100 μL containing $4 \times 10^7$ bacteria were harvested by centrifugation at 7000 g for 10 min. For kinetic experiments, 0.1% arabinose and 400 μM IPTG are added at $OD_{600}$ = 0.8 (t0), then after 20 min every 10 mins (for 100–130 mins) 100 μL from each culture were harvested by centrifugation to separate culture supernatant and bacterial pellet at 7000 g for 10 min. For double cysteine variants, 5 mM TCEP was added during the induction period (TCEP '+'). 1X SDS-loading buffer was added into the culture supernatants and heated at 95°C. The whole cells (pellet) were washed and resuspended in 100 μL of 50 mM Tris-HCl pH7.4, 200 mM NaCl 1X SDS-loading buffer, heated at 95°C for 10 min at 1400 rpm shaking. 10 μL fractions from supernatant and whole cells were analyzed on NuPAGE 4–12% gels (Invitrogen) with 1X MES SDS-PAGE running buffer for 35 min at 200 V (constant) and transferred to polyvinylidenedifluoride (PVDF) membrane via the iBlot system (Invitrogen). Anti-HIS-HRP, Anti-GroEL, Anti-mouse and rabbit IgG-HRP (Sigma) and Anti-MBP (NEB) antibodies were used for western blot analysis and imaged using an Image-Quant LAS 4000 imaging system (GE Healthcare). The respective amounts, and estimation of the secretion efficiency were determined by scanning densitometry of the CdiA-NT and CdiB protein bands using the ImageQuantTL software (GE Healthcare). For kinetic experiments, the secretion efficiency of CdiA-NT wild type is arbitrarily set to 100% at t130 min and compared with the secretion efficiency of CdiA-NT variants. For the DsbA experiment, MC4100 dsbA+ parent cells and MC4100 dsbA- cells (dsbA::cm) were transformed using pBAD-cdiB$^{Ab}$ D225C/F554C. All experiments were independently repeated at least three times from fresh transformations, and data were analyzed and presented using Prism 8 (GraphPad).

## Equilibration Molecular dynamics (MD) simulations

CHARMM-GUI membrane builder (*Jo et al., 2008*; *Wu et al., 2014*) was used to generate simulation systems. One copy of CdiB from *E. coli* or *A. baumannii* (referred to as CdiB$^{Ec}$ or CdiB$^{Ab}$, respectively) was inserted into its respective species-specific outer membrane. The outer leaflet of the *E. coli* membrane was composed of type 1 Lipid A and R1 core (*Wu et al., 2013*). The outer leaflet of the *A. baumannii* membrane was composed of type 1 and type 2 Lipid A in a 1:1 ratio and R1 core (*Fregolino et al., 2010*). The inner leaflets of both membranes were composed of PPPE, PVPG and PVCL with a ratio of 15:4:1 (*Vance and Vance, 2002*). All systems were solvated with TIP3P water (*Jorgensen et al., 1983*). The *E. coli* system size is 110 × 115×125 Å$^3$ and ~170000 atoms, while the *A. baumannii* system size is 115 × 125×125 Å$^3$ and ~200000 atoms. Visual Molecular Dynamics (VMD) was used to construct both systems (*Humphrey et al., 1996*). We ran six equilibrium simulations, three for CdiB$^{Ab}$ and three for CdiB$^{Ec}$, with NAMD for 500 ns each (*Phillips et al., 2005*). The lipid placement was the same for each replica. The force field used for all simulations is CHARMM36m (*Huang et al., 2017*). Langevin dynamics (damping constant ɣ = 1.0 ps$^{-1}$) was used to keep the temperature (310 K = 37°C) constant and an anisotropic Langevin piston barostat was adopted for constant pressure (one atm) (*Martyna et al., 1994*). The time step of the simulations is two fs. Bonded interactions and short-range (below the 12 Å cutoff) nonbonded interactions were

updated at every time step. The Particle-mesh Ewald (PME) method (*Darden et al., 1993*) was used for long-range interactions, calculated every other time step.

The potential of mean force (PMF) for extraction of the helix H1 from CdiB$^{Ab}$ and CdiB$^{Ec}$ β-barrels was calculated using replica-exchange umbrella sampling (REUS) (*Sugita et al., 2000*) and the weighted histogram analysis method (WHAM) (*Grossfield, 2017*; *Kumar et al., 1992*). A collective variable defining the distance between the $C_\alpha$ atoms of the helix and those of the barrel projected onto the membrane axis was constructed using the colvars module of NAMD (*Fiorin et al., 2013*). Steered molecular dynamics (SMD) (*Sotomayor and Schulten, 2007*) was used to generate starting states for each window. A total of 43–50 windows were used, covering a range from +2 to −40 Å (positive values are on the extracellular side and negative values on the periplasmic side compared to the crystal structure at 0 Å). For the four PMF calculations (two for CdiB$^{Ab}$ and two for CdiB$^{Ec}$), we ran 35–235 ns/window (1.7–10.1 μs per PMF). Exchange rates between the windows were between 0.1 and 0.5. A particularly difficult region to sample was around −4 to −12 Å for CdiB$^{Ab}$. To improve sampling in this region, we ran 16 additional windows, spaced every 0.5 Å, with a larger restraint for 10 ns each. While these windows helped with WHAM, it should be noted that they do not benefit from accelerated convergence intrinsic to REUS. Bootstrap error estimates were calculated using WHAM. Correlation times for each window, a necessary input for the error estimate, were determined from the integral of the normalized autocorrelation function (ACF), cut-off at the first time it reaches 0. These ACFs were determined using ACFCalculator (*Gaalswyk et al., 2016*) with inputs from additional 2-ns highly restrained (force constant of 10 kcal/mol·Å$^2$) simulations. The correlation time ranged from 20 ps to 140 ps and were smoothed using a three-point running average (*Hub et al., 2010*). PMFs are plotted in *Figure 7—figure supplement 2* with + / - 1 standard deviation.

To quantify H1's removal from the CdiB β-barrel, we ran SMD simulations. Because interactions with the linker region and POTRA domains are unpredictable in a non-equilibrium simulation, we deleted them prior to running SMD for both CdiB$^{Ab}$ and CdiB$^{Ec}$. In the SMD simulations, force was applied to the center of mass of H1 and its secondary structure was restrained to obtain the force profile of extraction under the assumption that H1 maintains its α-helical structure. H1 was pulled toward the periplasm at a constant speed (0.29 Å/ns) in all SMD simulations, which were 150–200 ns in length. Loop 2 (Leu260 to Ser266) was deleted in the 'ΔL2' system. Gly227 (β1) and Thr552 (β16) were substituted by cysteines to form a disulfide bond in the 'β1–β16' system. Three SMD replicas were run for each of these CdiB$^{Ab}$ variants at the same speed and length as for the WT systems.

To explore the different positions of H1 observed in each species' β-barrel, Targeted Molecular Dynamic (TMD) simulations were run. In the first TMD simulation starting from the crystal structure of CdiB$^{Ab}$, H1 was forced to adopt the same position relative to the β-barrel as that of CdiB$^{Ec}$ H1. In the second TMD simulation, the same procedure was applied to CdiB$^{Ec}$, matching its H1 to the position of that in the CdiB$^{Ab}$ structure. TMD was run for 30 ns, followed by 30 ns in which H1 was restrained to its new position, and finally 120 ns of free equilibration.

## Acknowledgements

We thank H Bernstein and X Wang for providing the MC4100 cells and for the stimulating discussions. This work was supported by the Intramural Research Program of the National Institutes of Health (NIH), National Institute of Diabetes and Digestive and Kidney Diseases (JG, IB, and SKB). ZZ, KL, and JCG are supported by a National Science Foundation CAREER award (MCB-1452464) and a National Institutes of Health award (R01-GM123169). Computational resources were provided through the Extreme Science and Engineering Discovery Environment (XSEDE; TG-MCB130173), which is supported by NSF grant number ACI-1548562. We thank the staff at GM/CA and SER-CAT beam lines (Advanced Photon Source, Argonne National Laboratory) and at Stanford Synchrotron Radiation Lightsource (SLAC National Accelerator Laboratory) for their assistance during data collection. Use of the Advanced Photon Source was supported by the US Department of Energy, Office of Science, Office of Basic Energy Sciences.

## Additional information

### Funding

| Funder | Grant reference number | Author |
|---|---|---|
| National Institute of Diabetes and Digestive and Kidney Diseases | Intramural Research Program | Jeremy Guerin<br>Istvan Botos<br>Susan K Buchanan |
| National Science Foundation | MCB-1452464 | Zijian Zhang<br>Karl Lundquist<br>James C Gumbart |
| National Institutes of Health | R01-GM123 | Zijian Zhang<br>Karl Lundquist<br>James C Gumbart |

The funders had no role in study design, data collection and interpretation, or the decision to submit the work for publication.

### Author contributions

Jeremy Guerin, Conceptualization, Formal analysis, Validation, Investigation, Writing - original draft, Writing - review and editing; Istvan Botos, Formal analysis, Investigation, Writing - review and editing; Zijian Zhang, Karl Lundquist, Formal analysis, Investigation; James C Gumbart, Resources, Supervision, Investigation, Writing - review and editing; Susan K Buchanan, Conceptualization, Resources, Supervision, Funding acquisition, Writing - original draft, Project administration, Writing - review and editing

### Author ORCIDs

Jeremy Guerin (iD) https://orcid.org/0000-0003-2622-040X
James C Gumbart (iD) http://orcid.org/0000-0002-1510-7842
Susan K Buchanan (iD) https://orcid.org/0000-0001-9657-7119

### Decision letter and Author response

Decision letter https://doi.org/10.7554/eLife.58100.sa1
Author response https://doi.org/10.7554/eLife.58100.sa2

## Additional files

### Supplementary files

• Supplementary file 1. Sequence alignment of TpsB transporters. The first two sequences correspond to CdiB$^{Ab}$ and CdiB$^{Ec}$ studied in this paper, where the names of strains are indicated. Amino acid sequences from CdiB to BcpB correspond to proteins involved in CDI mechanisms, whereas the last nine sequences (FhaC to LepB) represent other TpsB transporters. α-helices and β-strands are shown by cylinders and arrows respectively, from the structures of CdiB$^{Ab}$ (light teal) and CdiB$^{Ec}$ (pale yellow). The boundaries of POTRA domains are indicated in gray. Structural elements discussed in the text and figures are indicated in bold black, and important residues are marked with a star and label, in light teal for CdiB$^{Ab}$ and pale yellow for CdiB$^{Ec}$. Omp85/TpsB conserved residues on L6, β13, β1 and cysteines (involved in disulfide bond formation) are shown in red.

• Transparent reporting form

### Data availability

Diffraction data have been deposited in PDB under the accession code 6WIL and 6WIM. All data generated or analysed during this study are included in the manuscript and supporting files and videos.

The following datasets were generated:

| Author(s) | Year | Dataset title | Dataset URL | Database and Identifier |
|---|---|---|---|---|
| Guerin J, Botos I, Buchanan SK | 2020 | CdiB from Acinetobacter baumannii | https://www.rcsb.org/structure/6WIL | RCSB Protein Data Bank, 6WIL |
| Guerin J, Botos I, Buchanan SK | 2020 | CdiB from Escherichia coli | https://www.rcsb.org/structure/6WIM | RCSB Protein Data Bank, 6WIM |

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
