## [Decision Letter]

**Acceptance summary:**

This manuscript presents structures of two "CdiB" transporters, which serve as microbial defense systems, secreting substrate CdiA proteins to trigger toxic effects on competing microbes. To relate structure to function, the authors developed a functional assay and validated the role of the observed conformational change in the transport mechanism.

**Decision letter after peer review:**

Thank you for submitting your article "Structural insight into toxin secretion by contact dependent growth inhibition transporters" for consideration by *eLife*. Your article has been reviewed by four peer reviewers, including Merritt Maduke as the Reviewing Editors, Reviewer #1, and the evaluation has been overseen by and Olga Boudker as the Senior Editor. The following individuals involved in review of your submission have agreed to reveal their identity: Dukas Jurenas (Reviewer #3); Trevor Moraes (Reviewer #4).

The reviewers have discussed the reviews with one another, and the Reviewing Editor has drafted this decision to help you prepare a revised submission.

We would like to draw your attention to changes in our revision policy that we have made in response to COVID-19 (https://elifesciences.org/articles/57162). Specifically, when editors judge that a submitted work as a whole belongs in *eLife* but that some conclusions require a modest amount of additional new data, as they do with your paper, we are asking that if it is not feasible to do the experiments that the manuscript be revised to either limit claims to those supported by data in hand, or to explicitly state that the relevant conclusions require additional supporting data.

Summary:

In this manuscript, the authors report their findings on a type Vb two-partner secretion system, represented by CdIA and CdIB proteins from *Acinetobacter baumannii* and *Escherichia coli*. In this system, outer membrane protein CdIB is responsible for secretion of its substrate protein CdIA from periplasm to the extracellular environment, which then is delivered to a cell of a competing microbe, triggering its toxic effects on that cell. Thus, this system acts to inhibit growth of competing bacteria. Here, the structures of two CdiB transporters are presented. Previously, only one TpsB structure had been determined. The two new structures adopt two different conformations, one overlapping with that of the previous structure, and one new conformation involved a repositioning of a DxxG motif that is unique to the TpsB transporters. The writing in this manuscript is excellent; the Introduction clearly set up the background, original discovery of contact dependent inhibition, and the known biochemical/structural information about the type Vb secretion system. To test the functional relevance of predicted conformational change, the authors developed a secretion assay that measures secreted CdiA protein by Western blot, and they tested predictions via cross-linking, complementarity, and mutational analysis. The authors additionally perform MD simulations to evaluate energetics of removal of the H1 helix from the pore. The simulations support the main thrust of the paper, but there are major concerns that must be addressed. In summary, this manuscript presents the first structures of CdiB transporters, identifies conformational change in a critical structure motif, confirms the functional relevance of the observed conformational change, and uses MD simulations to lay groundwork for future experimental analysis of CdiB transport mechanism. Going forward, we look forward to follow-up work such as structures with bound substrates/substrate peptides, structures of mutants, and experimental (e.g. EPR/DEER) results on CdiB conformational dynamics. Overall, once issues with the MD simulations are addressed, the results presented in this manuscript will be of broad interest to scientists studying mechanisms of membrane proteins.

Experimental:

1) In Figure 6B, the authors show that the GGAG mutant increases apparent activity, which is ostensibly consistent with their model. However, it is feasible that the increase in CdiB(Ab) secretion is due to increased cell lysis (as occurs with the Ec homolog, Figure 4A) rather than to increased secretion activity. Therefore, it is important to show a control for this experiment. If an experiment is not possible at this time, then the paper should be revised to specify the caveat.

2) Figure 5B would benefit from having a WT CdIB protein as control (as was done in Figure 6A). Also, the statement that "CdiBAb was detected in the pellet after 20 minutes and remained at a constant level over the 100-minute time course, in the presence or absence of TCEP" does not seem correct, as the band intensity increased from 20min to 60min. Please correct this statement and if possible shown densitometry analysis on the bands to better quantify and describe the observed phenomenon.

Simulations:

The computational expert who reviewed the manuscript provided specific comments that we agree are essential to address. The reviewer's comments are as follows:

– Regarding the suggestion in subsection “Structural differences in helix H1” these two structures demonstrate mobility of the helix. How can we be certain that this difference in position is not a transporter dependent difference? This is exactly where the simulations could really help support this argument.

– Do the authors really see the CdiBAb simulation interconvert between the two beta1 configurations as stated in subsection “Link between DxxG conformation and position of H1 helix”, or do they only see one direction (unravelling)? I can't tell from the videos because it is too small. I would really like to see this analysis and any follow up, because while the removal of the helix in Figure 7 is very difficult to accurately capture, these changes are doable. Is there a lipid that plays a role in this happening? The authors do three simulations of each structure, is the lipid placement the same in each case or does that get reinitialized too?

– I want to give my general impressions of the helix removal simulations. I like that the authors used both umbrella sampling and SMD, but I must say that given the very large energies (and they may actually be very large) and significant contact between the helix and the barrel, I am completely unconvinced by this analysis as it stands. In agreement with the authors, I also worry about a number of things: the reaction coordinate, potential protein conformational changes in the barrel and loops, the action of other elements from the periplasm or extracellular spaces, and most importantly sampling. I am concerned that with a few microseconds of enhanced sampling, you are not going to answer this question adequately, and you may arrive at completely wrong conclusions. For instance, I am left with the impression that the authors believe that the helix unravels to exit, and while it may undergo conformational changes, I don't think it is going to look like your SMD where the entire thing is unravelled in the aqueous environment. My first reason to not think this is the case are the two structures – we see the helix move quite a bit, but it is a well-formed helix in both cases. Moreover, if you run the helix through DISOPRED, is there reason to think that it is marginally stable as an α-helix?

– Personally, I would really love to see a very careful PMF carried out between the two states that you currently have. Create a homology model of one structure using a template of the other, where needed, or do some other kind of targeting, and then really sample the move from one to the other. This alone would be really hard, and might take tens of microseconds, if not more, to get a converged data set, but there are some really good features of this approach:

– it is much more tractable,

– you have end points that you believe, and

– you will learn a lot about what it takes to move the helix in this barrel and how the helix and rest of the protein have to adapt to do it.

– With my 2 cents given, I am not opposed to including the current PMF and SMD (although, I wouldn't do it personally), but I really would like to see more simulations to feel better that you have reached some kind of convergence. This is outlined below. Moreover, I would include these as supplemental figures where I focused more on the very general big picture ideas of overall energies supporting one reaction coordinate/transformation over another, etc. But that is my opinion, and I don't want to tell you how to write your own paper (ok, I gave another 2 cents here).

PMF calculations

– How were the actual snapshots generated in the 50 windows? Did you use rigid rotation, or did you seed 1 window from the last? If you used rigid rotation/translation, what were the sterics like prior to each run? Did you minimize, etc? I think it would be most ideal to carry these windows out sequentially, where you start in the X-ray structure state, carry out a 20 ns simulation under restraints, and then pick back through the simulation to identify the snapshot that moved the farthest along your reaction coordinate to seed the next step.

– Figure 7C should have bootstrap error estimates, in my opinion. If it can't be done in Alan Grossfield's implementation (I can't remember) there are some Gromacs tools to do this and mBAR can do it too. It is important to down sample your data here in each bin, however, so that you don't use correlated snapshots. Given the complexity and size of this system, this is going to be very important to see. I would especially like to see in a reply the side by side comparisons of the figures here with any recalculated taking into account correlations.

– As far as I can tell, it looks like each PMF is made up of about 750-1,000 ns of aggregate data. While there are important questions about how this is seeded and the progress coordinate used, I would at least like to see 1 or 2 more of these PMF profiles for each of the two conditions generated using completely independent simulations to assess the role of sampling in the computed PMF values.

SMD simulations

– Two independent runs per each condition (each about 150-200 ns), but only 1 profile is plotted for each in Figure 7A and B. Did you plot the average of the two? Given the small amount of time used to generate these, I would certainly compute 3-5 more SMD runs per condition and make a plot with all of the runs so that we can assess the role of stochasticity and non-equilibrium pulling versus the differences in the protein and or mutation in the protein.

– Finally, do you see any correspondence between the structural conformations that come out of your PMF and the SMD? You might have mentioned this, but I didn't catch it. This kind of analysis would again make me feel better about the convergence, but since you don't seem to favor the helix exit as a helix, maybe you don't want to get into that.

The statistics on the simulations is inadequate.

[Editors' note: further revisions were suggested prior to acceptance, as described below.]

Thank you for re-submitting your article "Structural insight into toxin secretion by contact dependent growth inhibition transporters" for consideration by *eLife*. Your article has been re-reviewed by two peer reviewers, including Merritt Maduke as the Reviewing Editor and Reviewer #1, and the evaluation has been overseen by Olga Boudker as the Senior Editor.

The reviewers have discussed the reviews with one another and the Reviewing Editor has drafted this decision to help you prepare a revised submission.

Revisions:

It is great to see that PMF profile was extended and additional simulations run. The new PMF is very different from the previous ones. We would have liked to see more of an attempt to rationalize the distances and compare with this propertied structure. Are they at all close in a semiquantitative manner? For future studies, cross linking from this intermediate state would be ideal.

For the current study, the authors should qualify their PMF in the final text and include a statement noting that they got very different results for the same starting conditions, which highlights the high degree of uncertainty in these kinds of calculations, especially when the end point structures are not known.

---

## [Author Response]

Revisions for this paper:Experimental:1) In Figure 6B, the authors show that the GGAG mutant increases apparent activity, which is ostensibly consistent with their model. However, it is feasible that the increase in CdiB(Ab) secretion is due to increased cell lysis (as occurs with the Ec homolog, Figure 4A) rather than to increased secretion activity. Therefore, it is important to show a control for this experiment. If an experiment is not possible at this time, then the paper should be revised to specify the caveat.

GroEL and MBP controls have been added to Figure 6—figure supplement 1B for the CdiB Ab L2^GGAG^ variant. This control has been added in the results session.

2) Figure 5B would benefit from having a WT CdIB protein as control (as was done in Figure 6A). Also, the statement that "CdiBAb was detected in the pellet after 20 minutes and remained at a constant level over the 100-minute time course, in the presence or absence of TCEP" does not seem correct, as the band intensity increased from 20min to 60min. Please correct this statement and if possible shown densitometry analysis on the bands to better quantify and describe the observed phenomenon.

Figure 5B compares the presence vs. absence of a disulfide bond between β1 and β16. A control will be confusing since the controls are already presents in Figure 5A and Figure 6A. The statement about the detection of CdiB has been corrected in the text,

Simulations:The computational expert who reviewed the manuscript provided specific comments that the we agree are essential to address. The reviewer's comments are as follows:– Regarding the suggestion in subsection “Structural differences in helix H1” these two structures demonstrate mobility of the helix. How can we be certain that this difference in position is not a transporter dependent difference? This is exactly where the simulations could really help support this argument.

This is a great question. To address it, we used targeted MD to move H1 in each species’ barrel to the position of H1 in the other species (described in more detail below). We found that each is stable in its new position, indicated by the RMSD plots. Thus, we conclude that the precise orientation of H1 in the barrel is not a species-specific feature. See subsection “Structural differences in helix H1” and simulations have been added as Video 2. And RMSD plots analysis as Figure 2—figure supplement 1.

– Do the authors really see the CdiBAb simulation interconvert between the two beta1 configurations as stated in subsection “Link between DxxG conformation and position of H1 helix”, or do they only see one direction (unravelling)? I can't tell from the videos because it is too small.

We did observe interconversion between a loop and a β-strand for the top part of β1. This can be seen in Video 3.

I would really like to see this analysis and any follow up, because while the removal of the helix in Figure 7 is very difficult to accurately capture, these changes are doable. Is there a lipid that plays a role in this happening?

We observe that water penetrates into the membrane slightly to keep the DxxG motif solvated in our simulations (see Figure 7—figure supplement 1C). The water may facilitate this interconversion of secondary structure. A figure has been added as Figure 7—figure supplement 1C.

The authors do three simulations of each structure, is the lipid placement the same in each case or does that get reinitialized too?

The lipid placement is the same in each case. This is now noted in the manuscript in subsection “Equilibration Molecular dynamics (MD) simulations”.

– I want to give my general impressions of the helix removal simulations. I like that the authors used both umbrella sampling and SMD, but I must say that given the very large energies (and they may actually be very large) and significant contact between the helix and the barrel, I am completely unconvinced by this analysis as it stands. In agreement with the authors, I also worry about a number of things: the reaction coordinate, potential protein conformational changes in the barrel and loops, the action of other elements from the periplasm or extracellular spaces, and most importantly sampling. I am concerned that with a few microseconds of enhanced sampling, you are not going to answer this question adequately, and you may arrive at completely wrong conclusions. For instance, I am left with the impression that the authors believe that the helix unravels to exit, and while it may undergo conformational changes, I don't think it is going to look like your SMD where the entire thing is unravelled in the aqueous environment. My first reason to not think this is the case are the two structures – we see the helix move quite a bit, but it is a well-formed helix in both cases. Moreover, if you run the helix through DISOPRED, is there reason to think that it is marginally stable as an α-helix?

We concur with the reviewer that H1 is unlikely to completely unfold; we presented it to suggest that partial unfolding, though, might be relevant. We have observed this previously for TonB-dependent transporters, for which recent AFM work has shown that unfolding of the plug domain is part of the opening process (Hickman et al., Gating of TonB-dependent transporters by substrate-specific forced remodeling. Nat. Comm. 2017). We ran each H1 segment through DISOPRED as recommended and both are strongly predicted to be helical. In light of this, and the new PMF, we removed the SMD simulations that showed unfolding of H1. The new CdiB^Ec^ PMF has been added as Figure 7 (main figure). The new data are discussed in in subsection “Extraction of helix H1 from β-barrel lumen”.

– Personally, I would really love to see a very careful PMF carried out between the two states that you currently have. Create a homology model of one structure using a template of the other, where needed, or do some other kind of targeting, and then really sample the move from one to the other. This alone would be really hard, and might take tens of microseconds, if not more, to get a converged data set, but there are some really good features of this approach:– it is much more tractable,– you have end points that you believe, and– you will learn a lot about what it takes to move the helix in this barrel and how the helix and rest of the protein have to adapt to do it.

This is an intriguing idea. While we carefully considered doing a PMF, we ultimately decided it against it, primarily because the computational expense would be significant. Additionally, there was no obvious reaction coordinate, and using an approach like the string method, for example, would be even more expensive. As a compromise, we ran targeted MD to generate two new structures: (1) CdiB^Ab^ with its H1 in the position of that from CdiB^Ec^ and (2) CdiB^Ec^ with its H1 in the position of that from CdiB^Ab^. TMD was run for 30 ns, followed by 30 ns of H1 restrained to its new position, and then 120 ns of free equilibration. We present the RMSD for H1 compared to its original position and to the new one. We find that H1 in the two systems are stable in their new positions, suggesting that one orientation is not strictly favored over the other for either system. See subsection “Structural differences in helix H1” and TMD simulations have been added as Video 2. And RMSD plots analysis as Figure 2—figure supplement 1.

– With my 2 cents given, I am not opposed to including the current PMF and SMD (although, I wouldn't do it personally), but I really would like to see more simulations to feel better that you have reached some kind of convergence. This is outlined below. Moreover, I would include these as supplemental figures where I focused more on the very general big picture ideas of overall energies supporting one reaction coordinate/transformation over another, etc. But that is my opinion, and I don't want to tell you how to write your own paper (ok, I gave another 2 cents here).

We sincerely appreciate the reviewer’s candid thoughts. We ran more replicas for the SMD but moved them all to supplementary materials. Instead, Figure 7 in the main text is solely the new 10-μs PMF for CdiB^Ec^. The new CdiB^Ec^ PMF has been added as Figure 7 (main figure). SMD are now presented in Figure 7—figure supplement 2B (three replicas). To simplify the figure and clarify the message, the SMD analysis of CdiB^Ab^ S226E and CdiB^Ab^ R460G variants have been removed. The new data are discussed in subsection “Extraction of helix H1 from β-barrel lumen”.

PMF calculations– How were the actual snapshots generated in the 50 windows? Did you use rigid rotation, or did you seed 1 window from the last? If you used rigid rotation/translation, what were the sterics like prior to each run? Did you minimize, etc? I think it would be most ideal to carry these windows out sequentially, where you start in the X-ray structure state, carry out a 20 ns simulation under restraints, and then pick back through the simulation to identify the snapshot that moved the farthest along your reaction coordinate to seed the next step.

Snapshots were generated from the SMD simulations. The closest frame of the SMD trajectory within 0.15 Å of the target window center was selected to seed the REUS calculations. Some initial fraction of the REUS trajectory was then treated as equilibration (as much as 70 ns in the case of the longest one), and only data from after this point was used to determine the PMF. While not identical to the procedure suggested by the reviewer, it is similar. Furthermore, the additional PMF for each system was seeded from one of the new SMD simulations (described below).

– Figure 7C should have bootstrap error estimates, in my opinion. If it can't be done in Alan Grossfield's implementation (I can't remember) there are some Gromacs tools to do this and mBAR can do it too. It is important to down sample your data here in each bin, however, so that you don't use correlated snapshots. Given the complexity and size of this system, this is going to be very important to see. I would especially like to see in a reply the side by side comparisons of the figures here with any recalculated taking into account correlations.

Alan Grossfield’s WHAM code can generate bootstrap error estimates with an input of the correlation time in each window. To utilize this feature, we first ran additional highly restrained 2-ns simulations for each window with inputs taken from the REUS runs with the collective variable value output every time step. This data were then used along with the code ACFCalculator from Chris Rowley (https://github.com/RowleyGroup/ACFCalculator) to determine the autocorrelation function (ACF). The correlation time was taken as the integration of the normalized ACF, cut-off when it first reaches zero. Correlation times were then plotted for all windows and a three-point running average was taken to smooth it; this is following the procedure recommended for Gromacs in the g_wham source paper, Hub et al., 2010. Correlation times ranged from 20 ps to 140 ps. PMFs are now plotted with +/- 1 s.d. as determined by the bootstrap error analysis in WHAM.

Even in our first submission, trajectory data input to WHAM was sampled only every 20 ps, suggesting it is not significantly oversampled. Nonetheless, as WHAM does not use correlation times for the free-energy calculation, only the error analysis, we downsampled our data by a factor of 10 and recomputed the PMF from the second run of CdiB^Ec^. The difference between the two PMFs is at most 0.2 kcal/mol, and on average is 0.03 kcal/mol.

– As far as I can tell, it looks like each PMF is made up of about 750-1,000 ns of aggregate data. While there are important questions about how this is seeded and the progress coordinate used, I would at least like to see 1 or 2 more of these PMF profiles for each of the two conditions generated using completely independent simulations to assess the role of sampling in the computed PMF values.

From the repeated SMD simulations (described below), we seeded one new REUS simulation for each species. We also extended the two pre-existing REUS runs to 40 ns/window (CdiB^Ec^; 2000 ns total) and 35 ns/window (CdiB^Ab^; 1680 ns total).

In the case of CdiB^Ab^, the extended run only increased the value of the PMF at all points, peaking at ~90 kcal/mol instead of 70 kcal/mol originally. The second run of CdiB^Ab^ was lower at ~32 kcal/mol upon exit of the helix from the barrel (60 ns/window; 2640 ns total).

In the case of CdiB^Ec^, the results were more interesting. While extending the first run to 40 ns/window did little to change the PMF (< 2 kcal/mol difference on average), the second run was not nearly as high initially and flattened rather than continued rising as the helix was extracted from the barrel. To investigate this further, the REUS run was continued significantly longer, finally terminating at 235 ns/window (10.1 μs total), an order-of-magnitude more than previously run, when the change in successive 5 ns/window runs was minimal. From this PMF, we discovered a new minimum, one even lower in energy than the crystal structure minimum, at a point of intermediate extraction. The existence of an intermediate state during helix extraction from the barrel rationalizes earlier experimental data from one of us. In Guérin et al., 2014, pulsed-electron double-resonance spectroscopy (PELDOR) on a related two-partner-secretion protein, FhaC, revealed two distinct peaks in the distance distribution, even absent any substrate (see the first and third distance distributions in Figure 3 from Guérin et al., 2014).

In our PMF, the second minimum is separated from the crystal-structure conformation by ~7.5 kcal/mol and a barrier of ~13 kcal/mol; the fully extracted state is 8 kcal/mol higher than the second minimum and 1 kcal/mol higher than the crystal-structure conformation. We note, however, that these numbers will likely be shifted by the presence of the two POTRA domains, which were not included in these PMF calculations. Despite this approximation, the existence of an intermediate state, similar to that observed in the PELDOR distance distributions for the resting state of FhaC, gives us confidence that the second-run PMF of CdiB^Ec^ is most representative of the real extraction pathway for the helix. Thus, we have replaced Figure 7 in the paper with this PMF alone and relegated the other three to SI for completeness.

The new PMF have been added in Figure 7 and Figure 7—figure supplement 2A. The new data are discussed in subsection “Extraction of helix H1 from β-barrel lumen”; and methods has been added in Materials and methods.

SMD simulations– Two independent runs per each condition (each about 150-200 ns), but only 1 profile is plotted for each in Figure 7A and B. Did you plot the average of the two?

The two independent runs originally referred to the run with H1 unfolded vs. that with H1 forced to remain helical. We have now deleted the former. SMD are now presented in Figure 7—figure supplement 2B.

Given the small amount of time used to generate these, I would certainly compute 3-5 more SMD runs per condition and make a plot with all of the runs so that we can assess the role of stochasticity and non-equilibrium pulling versus the differences in the protein and or mutation in the protein.

We repeated each SMD twice, giving three in total. While some of the runs look roughly similar to one another, others show marked differences between them, e.g., AB wt and AB S-S bond. Inspection of the trajectories indicates that the cause is the formation of an ion-bridged interaction between two acidic residues, namely E4 on H1 and D224, D261, or E469 in the CdiB^Ab^ variants. D261 is removed in the ΔL2 mutant, although we still see an ion-bridged interaction between E4 and E469 in one out of three trajectories. We have now taken all SMD plots from the main text figures and moved them to Figure 7—figure supplement 2B. Example ion-bridged interactions are now shown in Figure 7—figure supplement 3.

– Finally, do you see any correspondence between the structural conformations that come out of your PMF and the SMD? You might have mentioned this, but I didn't catch it. This kind of analysis would again make me feel better about the convergence, but since you don't seem to favor the helix exit as a helix, maybe you don't want to get into that.The statistics on the simulations is inadequate.

Indeed, at this point, we think such a comparison would not be helpful. While we think the significantly improved PMF is a more realistic version of extraction of H1, the SMD simulations are exposing potential sticking points, namely the ion-bridged interactions.

[Editors' note: further revisions were suggested prior to acceptance, as described below.]

Revisions:It is great to see that PMF profile was extended and additional simulations run. The new PMF is very different from the previous ones. We would have liked to see more of an attempt to rationalize the distances and compare with this propertied structure. Are they at all close in a semiquantitative manner? For future studies, cross linking from this intermediate state would be ideal.

We agree with the reviewer that the comparison should ideally be quantitative. However, we decided that there are too many uncertainties on both the experimental and simulation sides to make even a semi-quantitative comparison. These uncertainties include (1) difference in proteins (FhaC in Guerin et al., 2014 and CdiB here), (2) presence of spin labels experimentally, which will notably affect the distances, (3) inability of the experiments to measure larger distances than what was presented (see Figure 4 in Guerin et al., 2014), and (4) lack of POTRA domains in the PMF calculations. Nonetheless, at least the directions of the changes are in agreement. Specifically, the distance between the extracellular side of the helix is farther from the extracellular loops and closer to the periplasmic loops in the intermediate states identified in both the experiments and the PMF.

To avoid overstating our claim, we have modified the text as follows:

“The existence of an intermediate state during helix extraction from the barrel is supported by earlier experimental data where Pulsed-electron double-resonance (PELDOR) spectroscopy revealed distinct peaks in the distance distribution on a related two-partner-secretion transporter, FhaC, even in the absence of any substrate (Guérin et al., 2014).”

For the current study, the authors should qualify their PMF in the final text and include a statement noting that they got very different results for the same starting conditions, which highlights the high degree of uncertainty in these kinds of calculations, especially when the end point structures are not known.

Based on this suggestion, we have added a qualifying sentence:

“We note that for computational efficiency, the POTRA domains were not present in the PMF calculations; along with the substrate, they may shift the relative energies of the fully embedded, partially embedded, and fully extracted H1 conformations. Also, the different PMFs obtained for different starting conditions illustrates the systematic uncertainty present in these calculations when run for moderate lengths (1-2 μs in total), complicated further by the unknown end-state structure.”